# Approximate Equivariance via Projection-Based Regularisation

**Torben Berndt** [1]  **Jan Stühmer** [1,2]

## Abstract

Equivariance is a powerful inductive bias in neural networks, improving generalisation and physical consistency. Recently, however, non-equivariant models have regained attention, due to their better runtime performance and imperfect symmetries that might arise in real-world applications. This has motivated the development of approximately equivariant models that strike a middle ground between respecting symmetries and fitting the data distribution. Existing approaches in this field either rely on sampling from a group, incurring a high sample complexity, or explicitly parameterise a model as a sum of an equivariant and non-equivariant network. This work instead approaches approximate equivariance via a projection-based regulariser which leverages a layer-wise orthogonal decomposition of a network's layers into equivariant and non-equivariant components. In contrast to existing methods, this penalises non-equivariance at an operator level across the full group orbit, rather than point-wise as in sample-based approaches. We present a mathematical framework for computing the non-equivariance penalty exactly and efficiently in both the spatial and spectral domains. In our experiments, our method is competitive with prior approximate-equivariance approaches in task performance, while achieving substantial runtime gains over sample-based regularisers. [1]

## 1. Introduction

Over the past few years, equivariance has been proven to be a powerful design principle for machine learning models across chemistry (Thomas et al., 2018; Satorras et al., 2021; Brandstetter et al., 2022; Hoogeboom et al., 2022; Xu et al., 2024), physics (Bogatskiy et al., 2020; Spinner et al., 2024; Brehmer et al., 2025), robotics (Hoang et al., 2025), and engineering (Toshev et al., 2023). Recently, however, there has been a shift back towards non-equivariant models, most prominently AlphaFold-3 (Abramson et al., 2024). Non-equivariant architectures often allow more flexible feature parameterisations and can be easier to optimise as the search is not restricted to an equivariant hypothesis class. This broader parameter space may enable the optimiser to find better minima than if it was confined to strictly equivariant models (Pertigkiozoglou et al., 2024). Moreover, many existing equivariant architectures rely on specialised tensor products to preserve symmetry (Weiler & Cesa, 2019; Brandstetter et al., 2022), which can be less efficient to compute on modern GPUs than dense matrix–vector operations.

At the same time, recent work demonstrates that equivariance remains a valuable inductive bias even at scale (Brehmer et al., 2024), and, for example, state-of-the-art molecular property prediction models continue to leverage it (Liao & Smidt, 2023; Liao et al., 2024; Fu et al., 2025). This motivates approaches that retain the benefits of equivariance without incurring its full constraints or computational costs. A common approach is to promote equivariance in otherwise non-equivariant architectures at the level of samples - for example via data augmentation, as in AlphaFold-3 (Abramson et al., 2024), or pointwise equivariance penalties (Bai et al., 2025). In this work, we take a different perspective and introduce *projection-based equivariance regularisation*, a framework based on first-principles which allows tuning equivariance into any neural architecture on the operator level, thereby directly affecting the model weights. Our primary contributions are: (i) We propose a theoretically-grounded approach to regularise general machine learning models towards exact equivariance. (ii) Making use of the orthogonal decompostion of functions into equivariant and non-equivariant components, we are able to penalise non-equivariance on an operator level over the whole group orbit. (iii) We show how to efficiently calculate the closed-form projection by working in the Fourier domain, allowing efficient regularisation for continuous groups such as $SO(n)$. (iv) We empirically show that our method is competitive with existing approximate-equivariance approaches in task performance, while offering

[1]Heidelberg Institute for Theoretical Studies, Heidelberg, Germany [2]IAR, Karlsruhe Institute of Technology, Karlsruhe, Germany.

*Proceedings of the 43rd International Conference on Machine Learning*, Seoul, South Korea. PMLR 306, 2026. Copyright 2026 by the author(s).

[1]The source code is available at https://github.com/hits-mli/approximate-equivariance-projection.

especially large runtime gains over sample-based regularisers.

## 1.1. Related Work

A growing body of work relaxes strictly equivariant architectures to better capture approximate or imperfect symmetries in data. Finzi et al. (2021) model departures from symmetry by adding a small non-equivariant "residual" pathway to an otherwise equivariant network. This however, leads to a double parameterisation of the equivariant part, as the residual pathway also models equivariant functions. Kim et al. (2023) extend this to mixed symmetries. Romero & Lohit (2022) introduce partial group convolutions that activate only on a subset of group elements. For discrete groups, Wang et al. (2022c) propose relaxed group convolutions, later extended by Wang et al. (2024) to expose symmetry-breaking mechanisms; Hofgard et al. (2024) further generalise this framework to continuous groups. Veefkind & Cesa (2024) introduce a learnable non-uniform measure over the group within steerable CNNs, yielding partially equivariant SCNNs whose degree of symmetry breaking is explicitly encoded in the learned measure. This method however only addresses the special case of steerable CNNs, while we propose an architecture-agnostic regulariser. Samudre et al. (2025) enforce approximate equivariance through group-matrix–structured convolutional layers with low displacement rank, so that symmetry and its controlled violation are encoded as proximity to the group-matrix manifold, leading to highly parameter-efficient CNNs for discrete groups. McNeela (2024) introduce Lie-algebra convolutions with a non-strict equivariance bias, and van der Ouderaa et al. (2022) relax translation equivariance using spatially non-stationary convolution kernels. On graphs, Huang et al. (2023) develop approximately automorphism-equivariant GNNs. A complementary line of work studies how to measure equivariance (or its violation) and use it in training objectives. A number of works penalise pointwise deviations from equivariance constraints using randomly sampled group transformations: Bai et al. (2025) use this strategy for artifact reduction in imaging, Kouzelis et al. (2025) for VAEs in generative modelling, and Zhong et al. (2023) for depth and normal prediction in images. Finally, assuming a model splits into equivariant and non-equivariant parts, Pertigkiozoglou et al. (2024) propose to phase out the non-equivariant component during training. Manolache et al. (2025) extend this idea by controlling the trade-off via constrained optimisation.

## 2. Background

**Notation.** For inner-product vector spaces $V$ and $V'$, we denote the *identity* on $V$ by $I_V$ and write $\mathrm{Hom}(V, V')$ for the vector space of linear maps $V \to V'$, with $\mathrm{End}(V) :=$ $\mathrm{Hom}(V, V)$. We write $T^* : V' \to V$ for the adjoint and $U(V) = \{T \in \mathrm{End}(V) : TT^* = T^*T = I_V\}$ for the unitary group. For a measure $\mu$ on $V$ and an inner product $\langle \cdot, \cdot \rangle_{V'}$ on $V'$, we use the $L^2(\mu)$ inner product on functions $S, T : V \to V'$ given by $\langle S, T \rangle_\mu := \int_V \langle S(v), T(v) \rangle_{V'} \, d\mu(v)$ with norm $\|T\|_\mu^2 = \langle T, T \rangle_\mu$. For $T \in \mathrm{Hom}(V, V')$, if $\mu$ is an isotropic Gaussian on $V$, then $\|T\|_\mu^2 = \int_V \|T(v)\|_{V'}^2 \, d\mu(v) = \mathrm{tr}(T^*T) =: \|T\|_{\mathrm{HS}}^2$, which in finite dimensions is equal to the Frobenius norm $\|T\|_F = \sqrt{\sum_{i,j} |T_{ij}|^2}$.

**Unitary representations.** Given a group $G$, a *unitary representation* is a homomorphism $\pi : G \to U(V_\pi)$ into the unitary operators on a Hilbert space $V_\pi$; we call the pair $(V_\pi, \pi)$ a $G$-module. Two representations $\pi : G \to U(V_\pi)$ and $\pi' : G \to U(V_{\pi'})$ are said to be *isomorphic* if there exists a unitary $U : V_\pi \to V_{\pi'}$ with $\pi(g) = U \pi'(g) U^{-1}$ for all $g \in G$. A representation is *irreducible* if it is not isomorphic to a direct sum of non-zero representations $\pi \oplus \pi'$ where $\pi \oplus \pi' : G \to U(V \oplus V')$ is defined by $(\pi \oplus \pi')(g)(v, v') = (\pi(g)v, \pi'(g)v')$.

**Haar measure.** Let $G$ be a compact group. The *Haar measure* $\lambda$ is the unique *bi-invariant* and *normalised* measure, i.e. for all Borel sets $E \subset G$ and every $g \in G$ we have $\lambda(gE) = \lambda(Eg) = \lambda(E)$, and $\lambda(G) = 1$. We can view the Haar measure as a uniform distribution over the group $G$. Indeed, if $G$ is discrete, the Haar measure becomes the discrete uniform measure with $\lambda(\{g\}) = \frac{1}{|G|}$ for all $g \in G$.

**Equivariance and $G$-smoothing.** Let $T : (V, \pi) \to (V', \pi')$ be a (bounded) linear map between $G$-modules. We say $T$ is *$G$-equivariant* if $T(\pi(g)v) = \pi'(g) T(v)$ for all $g \in G$, $v \in V$. If the action on $V'$ is trivial ($\pi'(g) = I_{V'}$), we call $T$ *invariant*. Averaging over $G$ yields the *$G$-smoothing (Reynolds) operator*

$$P(T) = \int_G \pi'(g)^* T \pi(g) \, d\lambda(g). \tag{1}$$

**Projection onto the equivariant subspace.** When $\pi, \pi'$ are unitary, $P$ is the orthogonal projector (with respect to the $L^2(\mu)$ inner product) from $\mathrm{Hom}(V, V')$ onto the closed subspace of $G$-equivariant linear maps (Elesedy & Zaidi, 2021). The following structural decomposition will be useful.

**Lemma 2.1** (Elesedy & Zaidi (2021), Lemma 1)**.** *Let $\mathcal{H} \subset \{(V, \pi) \to (V', \pi')\}$ be a function space that is closed under $P$ (i.e. $P(T) \in \mathcal{H}$ whenever $T \in \mathcal{H}$). Define*

$$S = \{T \in \mathcal{H} : T \text{ is } G\text{-equivariant}\}, \tag{2}$$
$$A = \ker P = \{T \in \mathcal{H} : P(T) = 0\}. \tag{3}$$

*Then $P$ is an orthogonal projection with range $S$ and kernel $A$, and hence $\mathcal{H} = S \oplus A$.*

```
Projection for finite groups
```

```
1  def project_finite(W, group, rho_in,
        ↪ rho_out):
2      W_proj = zeros_like(W)
3      for g in group:
4          W_proj += rho_out[g].conj().T @ W @
        ↪ rho_in[g]
5      return W_proj / len(group)
```

```
Projection for continuous groups
```

```
1  def project_continuous(K, irreps,
        ↪ spatial_axes):
2      K_hat = fftn(K, axes=spatial_axes)
3      for pi in irreps:
4          K_hat[pi] = mask_and_average(K_hat[
        ↪ pi])
5      return ifftn(K_hat, axes=spatial_axes)
```

*Figure 1.* Pseudo-code for the equivariant projection for finite (left) and continuous groups (right).

In particular, every $T \in \mathcal{H}$ orthogonally decomposes uniquely as $T = P(T) + \big(T - P(T)\big)$, where $P(T)$ is the $G$-equivariant component $S$ and $T - P(T) \in A$ is its $G$-anti-symmetric component. Moreover, we have the following:

**Corollary 2.2.** *A function $T : (V, \pi) \to (V', \pi')$ is $G$-equivariant if and only if $P(T) = T$.*

## 3. Equivariant Projection Regularisation

Motivated by these observations, we propose a simple framework for learning (approximately) equivariant models: Let $\mathcal{H}$ be a hypothesis class and $L_{\text{task}}(T)$ a task-specific loss function for $T \in \mathcal{H}$. We learn $T$ by solving

$$T^* \in \arg\inf_{T \in \mathcal{H}} \ L_{\text{task}}(T) \ + \ \lambda_G \left\| P(T) \right\|_\mu^2 \qquad (4)$$

$$+ \ \lambda_\perp \left\| T - P(T) \right\|_\mu^2, \quad (5)$$

where $\lambda_G, \lambda_\perp \geq 0$ are hyperparameters. Intuitively, increasing $\lambda_\perp$ penalises $\left\| T - P(T) \right\|_\mu^2$ more strongly, which encourages $P(T) = T$, steering the solution toward stronger equivariance according to Lemma 2.1. The hyperparameter $\lambda_G$ controls regularisation of the equivariant part. Since $P(T)$ and $T - P(T)$ are orthogonal components, we have $\left\| T \right\|_\mu^2 = \left\| P(T) \right\|_\mu^2 + \left\| T - P(T) \right\|_\mu^2$. Hence, if $T$ is linear and for $\lambda_G = \lambda_\perp$ this reduces to standard Frobenius weight regularisation $\lambda_G \left\| T \right\|_F^2$.

In what follows, we provide a theoretical justification for using $\left\| T - P(T) \right\|_\mu$ as a regulariser towards stronger equivariance. Recalling that $P(T)$ denotes the closest equivariant

operator to $T$, we show that the distance $\left\| T - P(T) \right\|_\mu$ is quantitatively equivalent to a natural measure of non-equivariance, the *equivariance defect*.

### 3.1. Bounding the Equivariance Error

**Definition 3.1** (Equivariance defect). Let $T$ be a function between $G$-modules with actions $\pi_{\text{in}}$ and $\pi_{\text{out}}$. The *equivariance defect* at $g \in G$ is

$$\Delta_g(T) \ \coloneqq \ \pi_{\text{out}}(g) \circ T \ - \ T \circ \pi_{\text{in}}(g), \qquad (6)$$

and the *worst-case defect* is

$$\mathcal{E}(T) \ \coloneqq \ \sup_{g \in G} \left\| \Delta_g(T) \right\|_\mu. \qquad (7)$$

By Lemma 2.1 (Elesedy & Zaidi, 2021), the quantity $\mathcal{E}(T)$ vanishes if and only if $T$ is $G$-equivariant. The next lemma shows that this defect is effectively controlled, up to constants, by the distance to the equivariant subspace measured by the projection $P$.

**Lemma 3.2.** *For every (Lipschitz) function $T$ between $G$-modules with unitary actions,*

$$\left\| T - P(T) \right\|_\mu \ \leq \ \mathcal{E}(T) \ \leq \ 2 \left\| T - P(T) \right\|_\mu. \quad (8)$$

*Proof.* See Appendix B.1 □

Lemma 3.2 shows that regularising by $\mathcal{E}(T)$ or by $\left\| T - P(T) \right\|$ is equivalent up to a factor of 2. Thus, minimising $\left\| T - P(T) \right\|$ minimises the worst-case defect.

In practice, $T$ will be some type of neural network architecture and is hence a composition of functions. The following bound decomposes the global defect of a network into per-layer defects, weighted by downstream Lipschitz constants.

**Lemma 3.3.** *Let $T = f_k \circ f_{k-1} \circ \cdots \circ f_1$ be a composition of Lipschitz maps between $G$-modules with unitary actions, and set $L_m \coloneqq \text{Lip}(f_m)$. Then*

$$\mathcal{E}(T) \ \leq \ \sum_{i=1}^{k} \Big( \prod_{m \neq i}^{k} L_m \Big) \mathcal{E}(f_i). \qquad (9)$$

*Proof.* See Appendix B.2 □

The bound above immediately yields the following corollary for standard feed-forward networks first shown by Kim et al. (2023).

**Corollary 3.4** (Kim et al. (2023)). *Let*

$$T \ = \ W^{(S)} \circ \sigma_{S-1} \circ W^{(S-1)} \circ \cdots \circ \sigma_1 \circ W^{(1)} \qquad (10)$$

$$T : L^2(G, V_{\text{in}}) \to L^2(G, V_{\text{out}}) \xrightarrow{\quad P_{\text{equiv}} \quad} T_{\text{equiv}} : L^2(G, V_{\text{in}}) \to L^2(G, V_{\text{out}})$$

Peter–Weyl $\quad$ ↻ $\quad$ Peter–Weyl

$$\widehat{T} = \{\widehat{T}(\pi, \sigma)\}_{\pi, \sigma \in \widehat{G}} \xrightarrow[\quad P: \{T_{\pi\sigma}\} \mapsto \{\delta_{\pi,\sigma} \operatorname{Av}_\pi(T_{\pi\pi})\} \quad]{} \widehat{T_{\text{equiv}}} = \bigoplus_{\pi \in \widehat{G}} (I_{V_\pi} \otimes W_\pi)$$

*Figure 2.* Commutative diagrams showing how to apply the projection operator in Fourier space. We zero out off-diagonals ($\pi \neq \sigma$) and average within each frequency block to obtain $I_{V_\pi} \otimes B_\pi$.

*be an S-layer network where each linear map $W^{(l)}$ acts between G-modules with unitary actions and each activation $\sigma_l$ is G-equivariant and Lipschitz. Then*

$$\mathcal{E}(T) \leq C \sum_{l=1}^{S} \left\| W^{(l)} - P(W^{(l)}) \right\|_F, \quad (11)$$

*for a constant $C > 0$ depending only on the operator norms of the $W^{(l)}$, the Lipschitz constants of the $\sigma_l$, and (when working on a bounded input domain) its radius.*

*Proof.* See Appendix B.3. □

### 3.2. Projection in Fourier Space

The previous section motivates the use of the norm of the projection operator as a regulariser. When the projection operator in Equation 1 is efficiently computable in the spatial domain, e.g., for small finite groups (see Section 4.3), this is straightforward; the algorithm in Figure 1 provides pseudocode for this case. However, in many applications, the group is large (for instance, uncountably infinite, as in $SO(n)$, the group of rotations about the origin in $\mathbb{R}^n$; see Sections 4.1 and 4.4). In such cases, the integral in Equation 1 rarely admits a closed-form solution.

We therefore switch to the spectral domain. We assume the following setup, which is in line with the geometric deep learning blueprint (Bronstein et al., 2021) that constructs equivariant networks as a composition of equivariant linear layers with equivariance-preserving non-linearities. Let $G$ be a compact group with normalised Haar measure $\lambda$, and consider linear maps $T : L^2(G) \to L^2(G)$ on the Hilbert space of square-integrable complex functions $L^2(G) = \{f : G \to \mathbb{C}\}$ with inner product

$$\langle f, h \rangle = \int_G f(g) \overline{h(g)} \, d\lambda(g). \quad (12)$$

We study equivariance with respect to the (left) regular representation $\tau : G \to U(L^2(G))$ defined by

$$(\tau(g)f)(x) = f(g^{-1}x), \qquad x, g \in G. \quad (13)$$

We denote by $\widehat{G}$ the set of equivalence classes of finite-dimensional irreducible representations of $G$ and call it

the *unitary dual* of $G$. Each $[\pi] \in \widehat{G}$ has a representative $\pi : G \to U(V_\pi)$ with $d_\pi = \dim V_\pi$. For $f \in L^2(G)$, we define the *(non-abelian) Fourier transform* as

$$\widehat{f}_\pi := \int_G f(g) \, \pi(g)^* \, d\lambda(g) \ \in \operatorname{End}(V_\pi). \quad (14)$$

By the Peter–Weyl theorem (Peter & Weyl, 1927), these coefficients form a complete spectral representation of $f$. More explicitly, the Fourier inversion formula reconstructs scalar functions as

$$f(g) = \sum_{\pi \in \widehat{G}} d_\pi \operatorname{tr}\left(\widehat{f}(\pi) \, \pi(g)\right), \quad (15)$$

with convergence in $L^2(G)$, and pointwise under the usual additional regularity assumptions. Thus, passing to Fourier space retains all information about functions.

Under the Peter–Weyl identification

$$L^2(G) \cong \bigoplus_{\pi \in \widehat{G}} V_\pi \otimes V_\pi^*, \quad (16)$$

a general linear operator $T : L^2(G) \to L^2(G)$ decomposes into frequency blocks

$$\widehat{T}_{\pi\sigma} : V_\sigma \otimes V_\sigma^* \to V_\pi \otimes V_\pi^*. \quad (17)$$

Equivalently, its action on Fourier coefficients can be written as

$$\widehat{(Tf)}(\pi) = \sum_{\sigma \in \widehat{G}} \widehat{T}_{\pi\sigma} \, \widehat{f}(\sigma). \quad (18)$$

A general operator may therefore mix different frequencies $\sigma \to \pi$. By Schur's lemma (Schur, 1905), equivariance imposes a block-diagonal structure on these blocks: it forbids mixing between inequivalent irreducible components and restricts the surviving diagonal blocks.

**Theorem 3.5** (Informal). *Equivariant linear maps are block-diagonal in the frequency domain (one block per irreducible representation). Hence, the projection onto equivariant subspaces acts by zeroing out all off-diagonal terms and averaging within.*

We schematically depict how the projection acts on weight matrices in Figure 3. Hence, whenever an efficient Fourier transform is available (e.g., on regular grids) or the model is already parameterised spectrally (e.g., eSEN (Fu et al., 2025)), imposing equivariance reduces to diagonalising the relevant linear operators in the spectral domain. The next section makes this more mathematically precise.

### 3.3. Equivariant Maps are Diagonal Across Frequencies

**Theorem 3.6.** *Let $T : L^2(G) \to L^2(G)$ be a linear function which is equivariant with respect to the (left) regular*

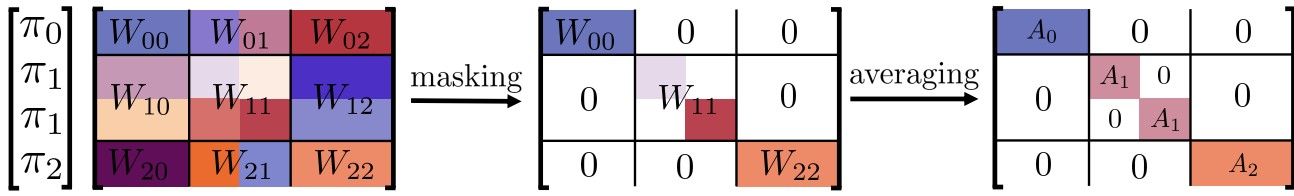

*Figure 3.* Projection of a linear map onto the equivariant subspace. The initial dense weight matrix mixes different irreducible grades $\pi_0, \pi_1, \pi_2$, with $\dim \pi_1 = 2$. The masking step removes all couplings between inequivalent grades, retaining only the isotypic blocks. Averaging then projects each retained block onto the equivariant form $I_{\pi_i} \otimes A_i$, where $A_i$ acts on the corresponding multiplicity space.

*representation, i.e. $\tau(g) \circ T = T \circ \tau(g)$ for all $g \in G$. Then $T$ decomposes as follows:*

$$\widehat{T} \cong \bigoplus_{\pi \in \widehat{G}} I_{V_\pi} \otimes B_\pi \qquad (19)$$

*for some $B_\pi \in End(V_\pi^*)$ (one for each $\pi$).*

*Proof.* See Appendix B.6. □

This means that an equivariant linear map $T$ does not mix between irreps; it is block-diagonal. We now show what this means for the projection of a general linear operator $T$.

**Corollary 3.7.** *Let $T : L^2(G) \to L^2(G)$ be linear and set $P_{equiv}(T)$ to be its equivariant projection. Then for each $[\pi] \in \widehat{G}$, there exists $B_\pi \in End(V_\pi^*)$ such that for all $f \in L^2(G)$,*

$$\widehat{P(T)(f)}(\pi) = \widehat{f}(\pi) \, B_\pi. \qquad (20)$$

### 3.4. Vector-valued Signals and Fiber-wise Projection

Thus far we treated scalar signals $f \in L^2(G)$. In many applications (e.g. steerable CNNs (Cohen & Welling, 2017), tensor fields) one works with vector-valued signals taking values in a finite-dimensional unitary $G$-module $(V, \rho)$. Define $L^2(G, V) \cong L^2(G) \otimes V$ with action

$$\big((\tau \otimes \rho)(g) f\big)(x) = \rho(g) \, f(g^{-1} x). \qquad (21)$$

More generally, for an operator $T : L^2(G, V_{\text{in}}) \to L^2(G, V_{\text{out}})$ we measure equivariance with respect to the pair of actions $\tau \otimes \rho_{\text{in}}$ (on the domain) and $\tau \otimes \rho_{\text{out}}$ (on the codomain), i.e.

$$(\tau \otimes \rho_{\text{out}})(g) \circ T = T \circ (\tau \otimes \rho_{\text{in}})(g) \qquad \forall g \in G. \ (22)$$

Under the identification

$$L^2(G, V) \cong \bigoplus_{\pi \in \widehat{G}} V_\pi \otimes V_\pi^* \otimes V,$$

a general operator $T : L^2(G, V_{\text{in}}) \to L^2(G, V_{\text{out}})$ has blocks

$$\widehat{T}_{\pi\sigma} : V_\sigma \otimes V_\sigma^* \otimes V_{\text{in}} \to V_\pi \otimes V_\pi^* \otimes V_{\text{out}}.$$

As before, the equivariant projection again removes all $\pi \neq \sigma$ blocks and projects each diagonal block onto the appropriate intertwiner space, leading to the following theorem. Details are provided in Appendix C.

**Theorem 3.8.** *Let $T : L^2(G, V_{\text{in}}) \to L^2(G, V_{\text{out}})$ be linear. Then the equivariant projection decomposes as*

$$\widehat{P_{\text{equiv}}(T)} \cong \bigoplus_{\pi \in \widehat{G}} \left( I_{V_\pi} \otimes W_\pi \right) \qquad (23)$$

*with*

$$W_\pi = \int_G \big(\pi(g)^* \otimes \rho_{\text{out}}(g)\big) \, \widehat{T}(\pi, \pi) \, \big(\pi(g) \otimes \rho_{\text{in}}(g)^{-1}\big) \, d\lambda(g).$$
$$(24)$$

*In particular, every equivariant $T$ is block-diagonal across frequencies and acts as the identity on $V_\pi$ and as an intertwiner on the fiber–multiplicity space $V_\pi^* \otimes V$.*

Hence, the equivariant projection can be computed efficiently in Fourier space. Given a linear map $T$, we (i) compute the Fourier transform of the matrix representation of $T$ to obtain the frequency blocks $\widehat{T}(\pi, \sigma)$; (ii) zero all off-diagonal blocks, setting $\widehat{T}(\pi, \sigma) \leftarrow 0$ for $\pi \neq \sigma$; (iii) for each $\pi$, project $\widehat{T}(\pi, \pi)$ onto $\text{Hom}_G(\pi^* \otimes \rho_{\text{in}}, \pi^* \otimes \rho_{\text{out}})$ using the averaging formula for $B_\pi$ above; and (iv) apply the inverse Fourier transform to obtain $P_{\text{equiv}}(T)$ in the spatial domain.

This procedure is illustrated by the commutative diagram in Figure 2, and a corresponding pseudo-code implementation is given in Algorithm 1 on the right.

### 3.5. Asymptotic Cost

We now want to briefly comment on the computational complexity of calculating the projection for both finite and continuous groups.

**Finite groups.** For finite groups we use Equation 1 directly. For a linear layer with weights $W \in \mathbb{C}^{d_{\text{out}} \times d_{\text{in}}}$ and $N_\ell = d_{\text{out}} d_{\text{in}}$ parameters, the projection evaluates $\pi_{\text{out}}(g)^* W \pi_{\text{in}}(g)$ for each $g \in G$, where $\pi_{\text{out}}(g)$, $\pi_{\text{in}}(g)$ are the representation matrices. Each step costs $O(d_{\text{out}}^2 d_{\text{in}}) + O(d_{\text{out}} d_{\text{in}}^2)$, which is $O(d_{\text{out}}^3)$ under $d_{\text{in}} \sim$

$d_{\text{out}}$. Since $N_\ell \sim d_{\text{out}}^2$, this is $O(N_\ell^{3/2})$ per group element, and $O\big(|G| N_\ell^{3/2}\big)$.

**Continuous groups.** For continuous groups, we use the Fourier-domain projection. If a model is parameterised spectrally, masking and averaging a weight matrix $W \in \mathbb{C}^{d_{\text{out}} \times d_{\text{in}}}$ cost $O(N_\ell)$. We note that this is precisely the regime of steerable CNNs, where kernels are parameterised directly in such blocks.

## 4. Experiments

In this section, we conduct three sets of experiments to demonstrate the feasibility and efficiency of our approach to learn (approximate) equivariance from data. For implementation details and information on hyperparameters, see Appendix D.[2]

### 4.1. Example: Learned $SO(2)$ Invariance

We first want to illustrate the approach in Section 3 on a simple toy problem (Figure 4). The task is binary classification on two point clouds in $\mathbb{R}^2$. Using polar coordinates $(r, \theta)$, we sample an inner disk-shaped cloud (blue, label $+1$), and the outer angular section of an annulus (red, label $-1$). We then train an approximately $SO(2)$-invariant MLP with the following structure on this dataset: We first project inputs $(x, y) \in \mathbb{R}^2$ onto circular harmonics up to degree $M$, adding $C$ radial channels via radial embedding functions, to obtain equivariant irreps features $H \in \mathbb{C}^{(2M+1) \times C_{\text{hid}}}$. We then apply two fully connected complex linear layers

$$L_i : \mathbb{C}^{(2M+1) \times C_{\text{hid}}} \to \mathbb{C}^{(2M+1) \times C_{\text{hid}}}, \tag{25}$$

followed by an $SO(2)$-equivariant tensor product. Lastly, we extract the invariant component and pass its real part through a final real-valued linear head $L_{\text{final}} : \mathbb{R}^{C_{\text{hid}}} \to \mathbb{R}$ to produce the scalar logit. For a more in-depth description of this architecture, see Appendix D.1.

In this setting, the projection onto the equivariant subspace reduces to masking. Let $W_i \in \mathbb{C}^{((2M+1)C) \times ((2M+1)C)}$ denote the flattened weight matrix of an intermediate linear layer. Define the mask $M \in \mathbb{R}^{((2M+1)C) \times ((2M+1)C)}$ by

$$M_{(m_1, c_1), (m_2, c_2)} = \delta_{m_1, m_2},$$

i.e., only blocks with matching harmonic order $m$ are kept. The projected weights are $P(W_i) = M \odot W_i$, where $\odot$ denotes elementwise multiplication. The overall objective is

$$L = L_{\text{task}} + \lambda_G \sum_i \|W_i\| + \lambda_\perp \sum_i \|W_i - M \odot W_i\|,$$

---

[2]Source code for reproducing the experiments will be released with the camera-ready version.

with $\lambda_G, \lambda_\perp \geq 0$ and $L_{\text{task}}$ the standard classification loss.

In Figure 4, we compare trained models across different values of $(\lambda_G, \lambda_\perp)$. From left to right, we first reduce $\lambda_G$ and then increase $\lambda_\perp$, enforcing progressively stronger invariance. For a full 2D grid for different combinations of $(\lambda_G, \lambda_\perp)$ see Figure 7 in Appendix D.1. As the regularisation intensifies, the decision boundary becomes increasingly $SO(2)$-invariant, confirming that the proposed projection-based regulariser effectively pushes the model toward invariance. Consistently, the empirical equivariance defect

$$\mathcal{E}_{\text{emp}}(T) = \sum_{k,l} \Big\| \rho_{\text{out}}(g_l) T(x_k) - T\big(\rho_{\text{in}}(g_l) x_k\big) \Big\| \tag{26}$$

with $k$ ranging over data samples and $g_l$ drawn as random rotations in $SO(2)$, decreases from left to right.

In a second experiment, we probe the behaviour of the regulariser when the target function departs from exact $SO(2)$-invariance by making the labels increasingly dependent on the polar angle. Starting from two concentric rings, we introduce an angular "wave" perturbation of amplitude $\sigma_\perp$ in the radial direction, such that for $\sigma_\perp = 0$ the data distribution is rotationally symmetric, whereas larger $\sigma_\perp$ produce interlocking rings (Figure 5 in appendix C). We train the approximately $SO(2)$-invariant network with projection-based regularisation alongside a plain MLP baseline on these datasets and compare both the learned decision boundaries and the empirical defect $\mathcal{E}_{\text{emp}}(T)$. As $\sigma_\perp$ increases, the regularised model departs from strict invariance only insofar as needed to fit the angularly perturbed rings. This illustrates how the projection penalty (even for constant values of $\lambda_\perp$) furnishes a tunable bias toward invariance that can be gradually traded off against fitting angle-dependent structure in the data. For a full grid, where we also vary the value of $\lambda_\perp$, see Figure 8 in Appendix D.1.

### 4.2. Imperfectly Symmetric Dynamical Systems

In this section, we follow the experimental design of Wang et al. (2022c) and evaluate our regulariser when applied to their relaxed group and steerable convolutional layers. Using PhiFlow (Holl & Thuerey, 2024), we generate $64 \times 64$ two-dimensional smoke advection–diffusion simulations with varied initial conditions under relaxed symmetries. Each network is trained to predict the velocity field one step ahead. To test generalisation, we consider two out-of-distribution settings. In the *Future* setting, models predict velocity fields at time steps that are absent from the training distribution, while remaining within spatial regions that were seen during training. In the *Domain* setting, we evaluate at the same time indices as training but at spatial locations that were not seen. The data are produced to break specific symmetries in a controlled way: for *translation*, we generate series for 35 distinct inflow positions and split

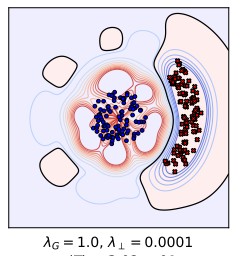 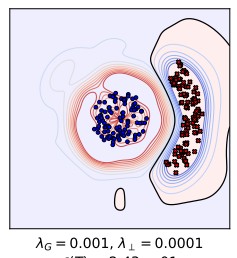 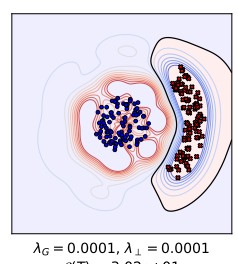 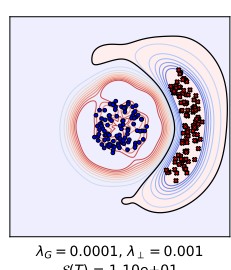 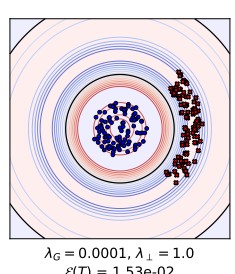

$\lambda_G = 1.0, \lambda_\perp = 0.0001$  $\quad$ $\lambda_G = 0.001, \lambda_\perp = 0.0001$  $\quad$ $\lambda_G = 0.0001, \lambda_\perp = 0.0001$  $\quad$ $\lambda_G = 0.0001, \lambda_\perp = 0.001$  $\quad$ $\lambda_G = 0.0001, \lambda_\perp = 1.0$
$\mathcal{E}(T) = 3.02\mathrm{e}{+}01$  $\quad$ $\mathcal{E}(T) = 2.43\mathrm{e}{+}01$  $\quad$ $\mathcal{E}(T) = 3.02\mathrm{e}{+}01$  $\quad$ $\mathcal{E}(T) = 1.10\mathrm{e}{+}01$  $\quad$ $\mathcal{E}(T) = 1.53\mathrm{e}{-}02$

*Figure 4.* Controlling the degree of learned $SO(2)$ invariance by tuning the parameters $\lambda_G$ and $\lambda_\perp$, which penalise the projections of the equivariant and non-equivariant components, respectively.

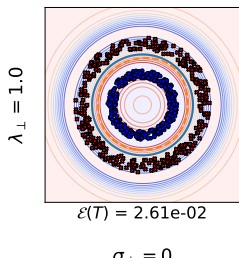 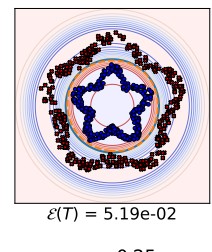 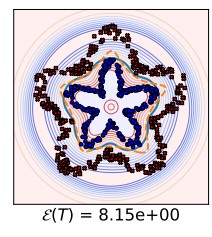 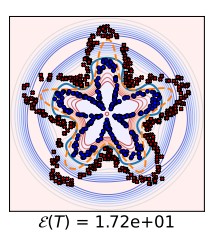 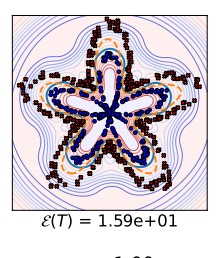

$\lambda_\perp = 1.0$

$\mathcal{E}(T) = 2.61\mathrm{e}{-}02$  $\quad$ $\mathcal{E}(T) = 5.19\mathrm{e}{-}02$  $\quad$ $\mathcal{E}(T) = 8.15\mathrm{e}{+}00$  $\quad$ $\mathcal{E}(T) = 1.72\mathrm{e}{+}01$  $\quad$ $\mathcal{E}(T) = 1.59\mathrm{e}{+}01$

$\sigma_\perp = 0$  $\quad$ $\sigma_\perp = 0.25$  $\quad$ $\sigma_\perp = 0.5$  $\quad$ $\sigma_\perp = 0.75$  $\quad$ $\sigma_\perp = 1.00$

*Figure 5.* Effect of increasing angular perturbation at fixed projection strength. Each panel shows the decision boundary and level sets of the approximately $SO(2)$-invariant network (blue) and an MLP (orange) trained with a fixed non-equivariant penalty $\lambda_\perp = 1.0$ on datasets with growing angular "wave" amplitude $\sigma_\perp$ (left to right). As $\sigma_\perp$ increases, the decision boundary becomes more angle-dependent and the learned classifier departs from perfect radial symmetry only where required to fit the data, while remaining nearly circular elsewhere. The empirical invariance defect $\mathcal{E}(T)$ for each setting is reported beneath the corresponding panel.

the domain horizontally into two subdomains with different buoyancy forces so that plumes diffuse at different rates across the interface; for *discrete rotation*, we simulate 40 combinations of inflow position and buoyancy, where the inflow pattern alone is symmetric under $90°$ rotations about the domain centre but a position-dependent buoyancy factor breaks rotational equivariance; and for *scaling*, we run 40 simulations with different time steps $\Delta t$ and spatial resolutions $\Delta x$ to disrupt scale equivariance. We compare the relaxed group convolutional networks (RGroup) and relaxed steerable CNNs (RSteer) introduced by Wang et al. (2022c) with several baselines: a standard CNN (Conv), an equivariant convolutional network (Equiv) (Weiler & Cesa, 2019; Sosnovik et al., 2020), Residual Pathway Priors (RPP) (Finzi et al., 2021), a locally connected network with an explicit equivariance penalty in the loss (CLNN) and Lift (Wang et al., 2022a). We indicate the addition of our regulariser with the suffix +Reg.

Across these settings, incorporating our regulariser preserves performance when approximate translation equivariance holds and delivers substantial improvements in the rotation and scaling regimes. In short, the penalty promotes the desired approximate equivariance where symmetry is only partially present, without degrading accuracy where the symmetry is already well aligned with the data.

### 4.3. CT-Scan Metal Artifact Reduction

We compare our approach with a sample-based equivariance penalty on metal artefact reduction (MAR) for CT scans. Metal implants introduce characteristic streaking artefacts that obscure clinically relevant structures. The task is to map a corrupted slice to its artefact-reduced counterpart. We use the AAPM CT-MAR Grand Challenge datasets (Haneda et al., 2025; AAPM, 2022), comprising 14,000 head and body CT slices with synthetic metal artefacts (Table 2 and Appendix D.3, Figure 9 for a visual comparison). The datasets were generated with the open-source CT simulation environment XCIST (Wu et al., 2022), using a hybrid data-simulation framework that combines publicly available clinical images (Yan et al., 2018; Goren et al., 2017) and virtual metal objects. Following Bai et al. (2025), we adapt three convolution-based architectures ACDNet (Wang et al., 2022b), DICDNet (Wang et al., 2021) and OSCNet (Wang et al., 2023) by encouraging rotation equivariance with respect to the discrete group $C_4$ (rotations by multiples of $90°$). We compare the unregularised baselines, the sample-based regulariser of Bai et al. (2025), and the same networks equipped with our projection-based regulariser. Additionally, we compare with Residual Pathway Priors (RPPs) (Finzi et al., 2021) and a train-then-project variant, in which we first train a non-equivariant model and then project its linear layers onto the equivariant subspace at test time using our projection operator.

*Table 1.* Results on three synthetic smoke-plume datasets exhibiting approximate symmetries. We report means and standard deviations of pixel-wise MSE over 5 random seeds. *Future* indicates that the test set occurs after the training period; *Domain* indicates that training and test sets come from different spatial regions. Adding our proposed *equivariance regulariser* (+Reg) consistently improves performance.

| Model | | Conv | Equiv | Rpp | CLCNN | Lift | RGroup | +Reg | RSteer | +Reg |
|---|---|---|---|---|---|---|---|---|---|---|
| Translation | Future | — | 0.94±0.02 | 0.92±0.01 | 0.92±0.01 | 0.87±0.03 | **0.71±0.01** | **0.72±0.01** | — | — |
| | Domain | — | 0.68±0.05 | 0.93±0.01 | 0.89±0.01 | 0.70±0.00 | **0.62±0.02** | **0.62±0.01** | — | — |
| Rotation | Future | 1.21±0.01 | 1.05±0.06 | 0.96±0.10 | 0.96±0.05 | 0.82±0.08 | 0.82±0.01 | 0.80±0.01 | 0.80±0.00 | **0.79±0.00** |
| | Domain | 1.10±0.05 | 0.76±0.02 | 0.83±0.01 | 0.84±0.10 | 0.68±0.09 | 0.73±0.02 | 0.67±0.01 | 0.67±0.01 | **0.58±0.00** |
| Scaling | Future | 0.83±0.01 | 0.75±0.03 | 0.81±0.09 | 1.03±0.01 | 0.85±0.01 | 0.80±0.01 | 0.81±0.00 | 0.70±0.01 | **0.62±0.01** |
| | Domain | 0.95±0.02 | 0.87±0.02 | 0.86±0.05 | 0.83±0.05 | 0.77±0.02 | 0.88±0.01 | 0.88±0.02 | 0.73±0.01 | **0.69±0.01** |

*Table 2.* CT metal artefact reduction on AAPM. We report PSNR/S-SIM, training throughput (batch sizes 4/12 where stable), inference throughput, epoch time, and peak memory; sample-based regularisation is limited to $\leq 4$, while baselines and ours scale to 12. Baselines: [1]Wang et al. (2022b), [2]Bai et al. (2025), [3]Finzi et al. (2021), [4]Wang et al. (2021), [5] Wang et al. (2023)

| Model | #params | Throughput (no./GPU-s) | | time (s) ↓ | Mem (GB) | AAPM | |
|---|---|---|---|---|---|---|---|
| | | Train ↑ | Inf. ↑ | | | PSNR ↑ | SSIM ↑ |
| ACDNet[1] | 4.2M | 4.90/5.16 | 8.40 | 1108 | 11.08 | 42.08 | 0.9559 |
| + sample-based[2] | 4.2M | 2.54/– | 8.38 | 2011 | 21.99 | 40.02 | **0.9623** |
| + test-time projection | 4.2M | – | – | – | – | 23.63 | 0.8384 |
| + RPP[3] | 6.9M | 3.49/4.14 | 5.37 | 1455 | 11.15 | 37.12 | 0.9413 |
| + projection-based (ours) | 4.2M | 4.25/4.99 | 7.44 | 1202 | 11.11 | **42.68** | 0.9620 |
| DICDNet[4] | 4.3M | 8.38/9.72 | 11.86 | 632 | 10.90 | 41.44 | 0.9468 |
| + sample-based[2] | 4.3M | 4.05/– | 10.15 | 1303 | 23.93 | 41.47 | 0.9464 |
| + test-time projection | 4.3M | – | – | – | – | 41.59 | 0.9602 |
| + RPP[3] | 6.6M | 3.10/6.10 | 6.93 | 1028 | 12.08 | 39.42 | 0.9481 |
| + projection-based (ours) | 4.3M | 5.77/7.82 | 10.11 | 782 | 12.05 | **41.52** | **0.9605** |
| OSCNet[5] | 4.3M | 8.59/9.86 | 12.00 | 624 | 10.37 | **42.36** | 0.9596 |
| + sample-based[2] | 4.3M | 4.05/– | 10.13 | 1304 | 23.93 | 41.50 | 0.9593 |
| + test-time projection | 4.3M | – | – | – | – | 41.37 | 0.9609 |
| + RPP[3] | 6.6M | 4.51/6.14 | 6.92 | 1016 | 12.08 | 39.45 | 0.9507 |
| + projection-based (ours) | 4.3M | 5.66/7.87 | 10.14 | 769 | 12.05 | 41.88 | **0.9612** |

For steerable CNN layers whose channels are organised into orientation groups of four, the layer-wise projection acting on a kernel $K \in \mathbb{R}^{C'_{\text{out}} \times C'_{\text{in}} \times 4 \times 4 \times s \times s}$ is

$$P_{\text{equiv}}(K) \;=\; \tfrac{1}{4} \sum_{r=0}^{3} S^r \left( \text{rot}_r K \right) S^{-r}, \qquad (27)$$

where $S$ is the $4 \times 4$ cyclic-shift matrix on orientation channels and $\text{rot}_r$ rotates the spatial kernel by $90° r$. For a derivation of this expression, see Appendix D.3.2.

In contrast, Bai et al. (2025) penalise a term that samples both a data point and a group element. For each sample $x$ they draw a random $r \in C_4$ and add

$$L_{\text{equiv}}(x, r) \;=\; \left\| S^r \, \text{rot}_r \, K(x) \;-\; K\big(S^r \, \text{rot}_r x\big) \right\|^2 \quad (28)$$

to the task loss. This requires an extra forward pass for each sampled rotation and each data sample, with asymptotic cost $O(N_{\text{samples}} \cdot \text{cost}_{\text{forward}})$ where $N_{\text{samples}}$ is the number of sampled group elements and $\text{cost}_{\text{forward}}$ is the cost of a single forward pass. By contrast, as derived in Section 3.5,

our projection-based regulariser $\|P_{\text{equiv}}(\cdot)\|$ incurs a cost that is linear in the number of parameters, does not sample rotations or data, introduces no extra forward passes, and has zero estimator variance.

In the fixed-batch setting, we use batch size 4 for all methods. In the max-feasible setting, the sample-based regulariser remains at batch size 4 (limited by the extra forward/activation memory), whereas the baselines and our projection-based regulariser scale to batch size 12 due to unchanged per-sample compute and memory. Our projection-based regulariser delivers competitive or superior reconstruction quality, surpassing the sample-based penalty in all metrics across all settings but one, and improving over the unregularised baselines in most cases. Owing to the extra forward pass in Equation 28, the sample-based approach is constrained to smaller batch sizes and lower throughput. Even under the fixed-batch protocol, its throughput is $42$–$47\%$ lower than ours; under the max-feasible protocol, the gap widens to $54$–$61\%$. These results indicate that projection-based regularisation achieves stronger $C_4$-equivariance with better hardware efficiency by avoiding per-sample group sampling. Similarly, due to the overparameterisation of the equivariant subspace, RPPs incur slower runtime during both training and inference and require more learnable parameters, and still underperform our approach in reconstruction quality.

### 4.4. MedMNIST

To evaluate our projection-based regulariser for *3D* steerable CNNs under *continuous* approximate symmetries, we follow the MedMNIST v2 (Yang et al., 2023) experiments of Veefkind & Cesa (2024). We consider the 3D classification tasks Nodule, Synapse, and Organ, which exhibit different degrees of (approximate) rotational symmetry. We compare a standard 3D CNN (CNN) to steerable CNNs (SCNNs) equivariant to $SO(3)$ and $O(3)$ (Weiler et al., 2018), RPPs (Finzi et al., 2021), and partially equivariant steerable CNNs (P-SCNN) from Veefkind & Cesa (2024). To apply our method, we use the ESCNN library (Cesa et al., 2022) to decompose each 3D convolutional kernel into equivariant and non-equivariant components, and regularise the two parts separately to encourage approximate $SO(3)$- or $O(3)$-equivariance. Table 3 reports test accuracy over 3 seeds,

*Table 3.* Test accuracies on MedMNIST datasets. Results for baselines (CNN, RPP, (P)-SCCN) are from Veefkind & Cesa (2024).

| Network Group | Partial Equivariance | Nodule | Synapse | Organ |
|---|---|---|---|---|
| CNN | N/A | 0.873 ±0.005 | 0.716 ±0.008 | 0.920 ±0.003 |
| SO(3) | None | 0.873 ±0.002 | 0.738 ±0.009 | 0.607 ±0.006 |
| | RPP | 0.801 ±0.003 | 0.695 ±0.037 | 0.936 ±0.002 |
| | P-SCNN | 0.871 ±0.001 | 0.770 ±0.030 | 0.902 ±0.006 |
| | ours | 0.862 ±0.018 | **0.774** ±0.025 | 0.933 ±0.000 |
| O(3) | None | 0.868 ±0.009 | 0.743 ±0.004 | 0.592 ±0.008 |
| | RPP | 0.810 ±0.013 | 0.722 ±0.023 | 0.940 ±0.006 |
| | P-SCNN | **0.873** ±0.008 | 0.769 ±0.005 | 0.905 ±0.004 |
| | ours | 0.860 ±0.020 | 0.773 ±0.025 | **0.943** ±0.005 |

*Table 4.* Computational efficiency on MedMNIST. We report the number of parameters, training and inference throughput, and epoch time.

| Model | #params | Throughput (samples/s) ↑ | | Epoch time (s) ↓ |
|---|---|---|---|---|
| | | Train | Inference | |
| CNN | 18.1M | 365.3 | 897.1 | 3.2 |
| SO(3)-SCNN | 0.1M | 196.7 | 666.7 | 5.9 |
| P-SCNN | 11.1M | 82.4 | 665.9 | 14.1 |
| RPP | 18.3M | 130.7 | 391.6 | 9.1 |
| ours | 2.0M | **638.2** | **5606.5** | **1.8** |

and Table 4 reports the corresponding parameter counts and runtime metrics.

For Nodule, most methods achieve comparable accuracy, with RPP as the main exception. In Synapse, (partial) rotational symmetry appears to be a helpful inductive bias: fully equivariant SCNNs improve over the CNN, while the best results are achieved by P-SCNN and the proposed projection-based method. By contrast, Organ seems to penalise strict equivariance: fully equivariant models underperform non-invariant alternatives, consistent with equivariance preventing left-right discrimination (see Appendix D.4.3). This asymmetry is further supported by the sensitivity study in Appendix D.4.2, where smaller values of $\lambda_\perp$ show to perform best on Organ, whereas Nodule and Synapse benefit from larger $\lambda_\perp$. The strongest performance here comes from the proposed method and RPP. At the same time, we note that our proposed method achieves higher training and inference throughput than the other considered models, and also attains the lowest epoch time.

Taken together, the results indicate that the proposed projection-based regulariser provides a favourable accuracy–efficiency trade-off. It remains robust across varying symmetry levels, retains the benefits of approximate equivariance, and avoids the substantial overhead of partial steerable CNNs and residual pathway priors.

## 5. Conclusion

In this work, we introduced projection-based regularisation - a theoretically grounded approach to learned equivariance which directly penalises model weights and regularises over the entire group instead of only point-wise, per-sample regularisation. For operators for which no closed-form solution of the projection can be computed efficiently in the spatial domain, we provide a general framework for computing the projection efficiently in Fourier space by masking. The experiments demonstrate that across synthetic and real-world experiments, covering both finite and continuous symmetry groups, the proposed approach improves both task performance and runtime.

**Limitations and future work.** The presented approach is "architecture-agnostic", in the sense that the general recipe of using the projection as a regulariser toward equivariance can be applied to any model. As show in Section 3, deriving this projection in closed form is straightforward for linear operators. In future work, we aim to study the projections of non-linearities and architecture-specific modules and use these to build architectures, which admit simple and efficient projections. Also, current experiments only evaluate the proposed method for relatively simple groups. In future work, we plan to extend this approach to more complex groups, for example with applications in material sciences.

## Impact Statement

This paper presents work aimed at advancing the field of learning under symmetries. There are many potential societal consequences of our work, none of which we feel must be specifically highlighted here.

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

$$f \in L^2(G) \xrightarrow{\quad P_{\text{inv}} \quad} f_{\text{inv}} \in L^2(G)$$

$$FT \Bigg\downarrow \Bigg\uparrow IFT \qquad \circlearrowleft \qquad FT \Bigg\downarrow \Bigg\uparrow IFT$$

$$\widehat{f} : \widehat{G} \to \bigcup_\pi U(V_\pi) \xrightarrow[\widehat{P_{\text{inv}}} = \delta_{\pi,\mathbf{1}}]{} \widehat{f}_{\text{inv}} = \widehat{f_{\text{inv}}} : \widehat{G} \to \bigcup_\pi U(V_\pi)$$

*Figure 6.* A commutative diagrams showing how to apply the projection operator in Fourier space. fro invariance, we keep only the trivial representation and discards all other frequencies.

## A. Projection in the Invariant Case

In this section, we show that an invariant function $f \in L^2(G)$ only has trivial non-zero Fourier coefficients.

**Lemma A.1.** *Let $f \in L^2(G)$ be left invariant with respect to the regular representation $\tau$, i.e. $f(hg) = f(g)$ for all $h, g \in G$. Then $\widehat{f}(\pi)$ is non-zero if and only if $\pi$ is the trivial representation $\mathbf{1} : g \mapsto I_{\mathbb{C}}$.*

*Proof.* See Appendix B.4. □

**Corollary A.2.** *Let $f \in L^2(G)$ be any function on $G$ and set $P_{\text{inv}}$ to be the invariant projection. Then $\widehat{P_{\text{inv}}(f)}(\pi) = \widehat{f}(\pi)\delta_{\pi,\mathbf{1}}$.*

*Proof.* See Appendix B.5 □

In Figure 6, we schematically depict how we can exploit the simple structure of the projection in the spectral domain $\widehat{P_{\text{inv}}}$ to efficiently calculate the smoothing operator $P_{\text{inv}}$.

## B. Proofs in Section 3

### B.1. Proof of Lemma 3.2

*Proof of Lemma 3.2.* By definition of $P$,

$$T - P(T) = \int_G \pi_{\text{out}}(g)^* \big( \pi_{\text{out}}(g) \circ T - T \circ \pi_{\text{in}}(g) \big) \, d\lambda(g) = \int_G \pi_{\text{out}}(g)^* \Delta_g(T) \, d\lambda(g). \tag{29}$$

Pre-/post-composition with the unitaries $\pi_{\text{out}}(g)^*$ preserves the Hilert-Schmidt norm, and the norm of an average is at most the average of the norm. Hence

$$\|T - P(T)\|_\mu \leq \int_G \|\Delta_g(T)\|_\mu \, d\lambda(g) \leq \sup_{g \in G} \|\Delta_g(T)\|_\mu = \mathbb{E}(T), \tag{30}$$

giving the lower bound. For the upper bound, note that $P(T)$ is $G$-equivariant, and therefore,

$$\Delta_g(T) = \pi_{\text{out}}(g) \big( T - P(T) \big) - \big( T - P(T) \big) \pi_{\text{in}}(g). \tag{31}$$

Taking norms and using that $\pi_{\text{in/out}}(g)$ are unitaries,

$$\|\Delta_g(T)\|_\mu \leq \|T - P(T)\|_\mu + \|T - P(T)\|_\mu = 2\|T - P(T)\|_\mu. \tag{32}$$

Finally, take the supremum over $g \in G$ to obtain $\mathbb{E}(T) \leq 2\|T - P(T)\|_\mu$. □

### B.2. Proof of Lemma 3.3

*Proof of Lemma 3.3.* For any composable maps $A, B$, the equivariance defect satisfies the chain rule

$$\Delta_g(A \circ B) = (\Delta_g A) \circ B + A \circ (\Delta_g B). \tag{33}$$

Applying this repeatedly to $f_k \circ \cdots \circ f_1$ yields the telescoping identity

$$\Delta_g(T) = \sum_{i=1}^{k} \left( f_k \circ \cdots \circ f_{i+1} \right) \circ \Delta_g(f_i) \circ \left( f_{i-1} \circ \cdots \circ f_1 \right). \tag{34}$$

Taking norms and using $\| f_{i+1} \circ \Delta_g(f_i) \|_\mu = \| f_{i+1} \circ (\pi_{out}(g) \circ f_i - f_i \circ \pi_{in}(g)) \|_\mu \leq \mathrm{Lip}(f_{i+1}) \| \Delta_g(f_i) \|_\mu$ iteratively together with $\mathrm{Lip}(f_j) = L_j$ to obtain

$$\| \Delta_g(T) \|_\mu \leq \sum_{i=1}^{k} \left( \prod_{m=i+1}^{k} L_m \right) \| \Delta_g(f_i) \|_\mu \left( \prod_{m=1}^{i-1} L_m \right). \tag{35}$$

Finally, take $\sup_{g \in G}$ on both sides and note that $\mathbb{E}(T) = \sup_g \| \Delta_g(T) \|$ and $\mathbb{E}(f_i) = \sup_g \| \Delta_g(f_i) \|$ to obtain the stated bound. $\qquad \square$

## B.3. Proof of Corollary 3.4

*Proof of Corollary 3.4.* To fit into the framework of Lemma 3.3, we choose

$$f_{2k-1} := W^{(k)}, \qquad f_{2k} := \sigma_k,$$

so that

$$L_{2k-1} := \| W^{(k)} \|_F, \quad L_{2k} := \mathrm{Lip}(\sigma_k),$$

for $k = 1, \ldots, 2S - 1$. By construction $E(\sigma_k) = 0$ for all $k$, hence $E(f_{2k}) = 0$. Plugging this into Equation 9 and noting that the even indices do not contribute, we obtain

$$E(T) \leq \sum_{k=1}^{S} \left( \prod_{m \neq 2k-1} L_m \right) E(W^{(k)}). \tag{36}$$

Now note that the product over $m \neq 2k - 1$ contains

- all activation Lipschitz constants $L_{2j} = \mathrm{Lip}(\sigma_j)$, $j = 1, \ldots, 2S - 1$,

- all weight norms $L_{2r-1} = \| W^{(r)} \|$ with $r \neq k$.

Thus

$$\prod_{m \neq 2k-1} L_m = \left( \prod_{j=1}^{S-1} \mathrm{Lip}(\sigma_j) \right) \left( \prod_{\substack{r=1 \\ r \neq k}}^{S} \| W^{(r)} \| \right),$$

and Equation 36 becomes

$$E(T) \leq \left( \prod_{j=1}^{S-1} \mathrm{Lip}(\sigma_j) \right) \sum_{k=1}^{S} \left( \prod_{\substack{r=1 \\ r \neq k}}^{S} \| W^{(r)} \| \right) E(W^{(k)}). \tag{37}$$

Next use Lemma 3.2, which states that for each linear layer

$$E(W^{(k)}) \leq 2 \| W^{(k)} - P(W^{(k)}) \|_F.$$

Substituting this into Equation 37 yields

$$E(T) \leq 2 \left( \prod_{j=1}^{S-1} \mathrm{Lip}(\sigma_j) \right) \sum_{k=1}^{S} \left( \prod_{\substack{r=1 \\ r \neq k}}^{S} \| W^{(r)} \|_F \right) \| W^{(k)} - P(W^{(k)}) \|_F. \tag{38}$$

Define

$$C := 2\left(\prod_{j=1}^{S-1} \mathrm{Lip}(\sigma_j)\right) \max_{1 \le k \le S} \prod_{\substack{r=1 \\ r \neq k}}^{S} \|W^{(r)}\|_F. \tag{39}$$

Then, for every $k$,

$$2\left(\prod_{j=1}^{S-1} \mathrm{Lip}(\sigma_j)\right) \prod_{\substack{r=1 \\ r \neq k}}^{S} \|W^{(r)}\| \ \le\ C,$$

and Equation 38 implies

$$E(T) \ \le\ C \sum_{k=1}^{S} \|W^{(k)} - P(W^{(k)})\|_F.$$

This is exactly Eq. (9), with the dependence of $C$ on the norms $\|W^{(k)}\|$ and Lipschitz constants $\mathrm{Lip}(\sigma_j)$ made explicit in Equation 39. $\qquad\square$

### B.4. Proof of Lemma A.1

*Proof of Lemma A.1.* We define the invariance operator of a function $f \in L^2(G)$ as

$$f_{\mathrm{inv}}(g) = \int_G f(hg)\, d\lambda(h) \tag{40}$$

The Fourier coefficients of this are

$$\widehat{f_{\mathrm{inv}}}(\pi) = \int_G f_{\mathrm{inv}}(g)\, \pi(g)^*\, d\lambda(g) \tag{41}$$

$$= \int_G \left(\int_G f(hg)\, d\lambda(h)\right) \pi(g)^*\, d\lambda(g) \tag{42}$$

$$= \int_G f(x) \left(\int_G \pi(h^{-1}x)^*\, d\lambda(h)\right) d\lambda(x) \quad \text{substituting } x = hg \implies g = h^{-1}x \tag{43}$$

$$= \int_G f(x) \left(\int_G \pi(h^{-1})^*\, d\lambda(h)\right) \pi(x)^*\, d\lambda(x) \tag{44}$$

$$= \int_G f(x) \left(\int_G \pi(h)^*\, d\lambda(h)\right) \pi(x)^*\, d\lambda(x). \quad \text{invariance of Haar measure} \tag{45}$$

Define $A_\pi := \int_G \pi(h)^*\, d\lambda(h) \in \mathrm{End}(V_\pi)$. Note that $A_\pi$ is $\pi$-equivariant; indeed, for all $g \in G$,

$$\pi(g)\, A_\pi = \int_G \pi(g)\, \pi(h)^*\, d\lambda(h) \tag{46}$$

$$= \int_G \pi(gh^{-1})\, d\lambda(h) \tag{47}$$

$$= \int_G \pi(k)^* \pi(g)\, d\lambda(k) \qquad \text{substituting } k = ghg^{-1} \implies gh^{-1} = k^{-1}g \tag{48}$$

$$= A_\pi\, \pi(g), \tag{49}$$

Hence by Schur's lemma (since $\pi$ is irreducible), we have

$$A_\pi \in \mathrm{End}G(V_\pi) \ \cong\ \{\lambda I : \lambda \in \mathbb{C}\}.$$

So $A_\pi = \lambda I$ for some $\lambda \in \mathbb{C}$.

Now,

$$\mathrm{tr}\, A_\pi = \int_G \mathrm{tr}\big(\pi(h)^*\big)\, d\lambda(h) = \int_G \overline{\chi_\pi(h)}\, d\lambda(h) \ = \ \overline{\int_G \chi_\pi(h)\, d\lambda(h)}. \tag{50}$$

But the characters $\chi_\pi$ are orthonormal, so denoting the trivial representation $g \mapsto 1$ by $\mathbf{1}$, i.e. have $\chi_{\mathbf{1}}(g) = 1$ for all $g$, we have

$$\int_G \chi_\pi(g) \, d\lambda(g) = \int_G \chi_\pi(g) \, \overline{\chi_{\mathbf{1}}(g)} \, d\lambda(g) = \langle \chi_\pi(g), \chi_{\mathbf{1}}(g) \rangle_{L^2(G)} = \delta_{\pi,\mathbf{1}}. \tag{51}$$

Finally, this gives

$$d_\pi \lambda \;=\; \operatorname{tr} A_\pi \;=\; \implies \; \lambda \;=\; \frac{\delta_{\pi,\mathbf{1}}}{d_\pi} = \begin{cases} 0, & \pi \neq \mathbf{1}, \\ \frac{1}{d_\pi}, & \pi = \mathbf{1}. \end{cases} \tag{52}$$

Substituting this into the above yields

$$\widehat{f_{\mathrm{inv}}}(\pi) = \frac{1}{d_\pi} \widehat{f}(\pi) \delta_{\pi,\mathbf{1}}. \tag{53}$$

$\square$

### B.5. Proof of Corollary A.2

*Proof of Corollary A.2.* Since $P_{\mathrm{inv}}$ is a projection onto the $G$-invariant subspace, $P_{\mathrm{inv}}(f)$ is always invariant. Hence, by Lemma A.1, $\widehat{P_{\mathrm{inv}}(f)}(\pi)$ is zero for all $\pi \neq \mathbf{1}$. Now note that by invariance, $P_{\mathrm{inv}}(f)(g) = c$ for all $g \in G$ for some $c \in \mathbb{C}$. We then calculate

$$\widehat{P_{\mathrm{inv}}(f)}(\mathbf{1}) = \int_G P_{\mathrm{inv}}(f)(g) \, \mathbf{1}(g)^* \, d\lambda(g) = \int_G P_{\mathrm{inv}}(f)(g) \, d\lambda(g) = \int_G c \, d\lambda(g) = c. \tag{54}$$

At the same time

$$\widehat{f}(\mathbf{1}) = \int_G f(g) \, \mathbf{1}(g)^* d\lambda(g) = \int_G f(g) d\lambda(g) = c. \tag{55}$$

which concludes the proof. $\square$

### B.6. Proof of Theorem 3.6

*Proof of Theorem 3.6.* Recall that by Peter–Weyl we have an isometric $G$-equivariant decomposition

$$L^2(G) \;\cong\; \bigoplus_{\pi \in \widehat{G}} V_\pi \otimes V_\pi^*,$$

where the left regular representation acts as $\tau(g) \cong \bigoplus_\pi \pi(g) \otimes I_{V_\pi^*}$. Let $T : L^2(G) \to L^2(G)$ be linear and $\tau$-equivariant, i.e. $\tau(g) \circ T = T \circ \tau(g)$ for all $g \in G$. Write the block-matrix of $T$ in this decomposition as $\widehat{T} = \{T_{\pi\sigma}\}_{\pi,\sigma \in \widehat{G}}$ with

$$T_{\pi\sigma} \; : \; V_\sigma \otimes V_\sigma^* \;\longrightarrow\; V_\pi \otimes V_\pi^*.$$

Equivariance implies, for all $g \in G$,

$$(\pi(g) \otimes I) \, T_{\pi\sigma} \;=\; T_{\pi\sigma} \, (\sigma(g) \otimes I). \tag{56}$$

Thus $T_{\pi\sigma}$ is an intertwiner from $\sigma$ to $\pi$ (acting on the first tensor factor). By Schur's lemma, $T_{\pi\sigma} = 0$ whenever $\pi \not\cong \sigma$. Hence $\widehat{T}$ is block-diagonal:

$$\widehat{T} \;\cong\; \bigoplus_{\pi \in \widehat{G}} T_{\pi\pi}, \qquad T_{\pi\pi} \in \operatorname{End}(V_\pi \otimes V_\pi^*),$$

and Equation 56 reduces to

$$(\pi(g) \otimes I) \, T_{\pi\pi} \;=\; T_{\pi\pi} \, (\pi(g) \otimes I) \qquad \forall g \in G. \tag{57}$$

We now identify all endomorphisms satisfying Equation 57. Consider the canonical vector space isomorphism

$$\Phi : \; V_\pi \otimes V_\pi^* \xrightarrow{\;\cong\;} \operatorname{End}(V_\pi), \qquad \Phi(u \otimes \varphi)(v) := \varphi(v) \, u.$$

A direct check shows that under $\Phi$, the action $\pi(g) \otimes I$ corresponds to left multiplication on $\mathrm{End}(V_\pi)$:

$$\Phi\big((\pi(g) \otimes I)(w)\big) \;=\; \pi(g)\,\Phi(w) \qquad \forall w \in V_\pi \otimes V_\pi^*.$$

Define $\widetilde{T}_\pi := \Phi \circ T_{\pi\pi} \circ \Phi^{-1} \in \mathrm{End}(\mathrm{End}(V_\pi))$. Then Equation 57 is equivalent to

$$\widetilde{T}_\pi(\pi(g)X) \;=\; \pi(g)\widetilde{T}_\pi(X) \qquad \forall g \in G,\ \forall X \in \mathrm{End}(V_\pi). \tag{58}$$

Let $B := \widetilde{T}_\pi(I_{V_\pi}) \in \mathrm{End}(V_\pi)$. For any $g \in G$, applying Equation 58 to $X = I$ gives $\widetilde{T}_\pi(\pi(g)) = \pi(g)B$. By linearity, the same identity holds for all $X$ in the linear span of $\{\pi(g) : g \in G\}$.

Since $\pi$ is irreducible, the span of $\{\pi(g)\}$ is all of $\mathrm{End}(V_\pi)$ (Burnside's theorem). Therefore, for every $X \in \mathrm{End}(V_\pi)$ we have

$$\widetilde{T}_\pi(X) \;=\; X\,B,$$

i.e. $\widetilde{T}_\pi$ is right multiplication by $B$. Transporting back through $\Phi$ shows that

$$T_{\pi\pi} \;=\; I_{V_\pi} \otimes B_\pi$$

for some $B_\pi \in \mathrm{End}(V_\pi^*)$ (canonically corresponding to $B$ on the dual space). Hence

$$\widehat{T} \;\cong\; \bigoplus_{\pi \in \widehat{G}} \big(I_{V_\pi} \otimes B_\pi\big),$$

which proves the claimed block structure. $\qquad\square$

## C. Details on Vector-valued Signals

**Fourier description.**   Peter–Weyl yields the unitary decomposition

$$L^2(G) \;\cong\; \bigoplus_{\pi \in \widehat{G}} V_\pi \otimes V_\pi^*, \qquad L^2(G, V) \;\cong\; \bigoplus_{\pi \in \widehat{G}} V_\pi \otimes \big(V_\pi^* \otimes V\big),$$

where $G$ acts by $\pi$ on the first tensor factor and trivially on $V_\pi^*$, while the fiber transforms by $\rho$. Accordingly, any bounded linear map $T : L^2(G, V_{\mathrm{in}}) \to L^2(G, V_{\mathrm{out}})$ admits a block form

$$\widehat{T} \;\cong\; \big(\widehat{T}(\pi, \sigma)\big)_{\pi, \sigma \in \widehat{G}}, \qquad \widehat{T}(\pi, \sigma) : V_\sigma \otimes \big(V_\sigma^* \otimes V_{\mathrm{in}}\big) \longrightarrow V_\pi \otimes \big(V_\pi^* \otimes V_{\mathrm{out}}\big).$$

Averaging annihilates all off-diagonal $(\pi \neq \sigma)$ blocks and, on each frequency $\pi$, orthogonally projects $\widehat{T}(\pi, \pi)$ onto the intertwiner space $\mathrm{Hom}_G\big(\pi^* \otimes \rho_{\mathrm{in}},\, \pi^* \otimes \rho_{\mathrm{out}}\big)$.

**Theorem C.1** (Theorem 3.8 restated). *Let $T : L^2(G, V_{\mathrm{in}}) \to L^2(G, V_{\mathrm{out}})$ be linear. Then*

$$\widehat{P_{\mathrm{equiv}}(T)} \;\cong\; \bigoplus_{\pi \in \widehat{G}} \big(I_{V_\pi} \otimes B_\pi\big), \tag{59}$$

$$B_\pi \;=\; \int_G \big(\pi(g)^* \otimes \rho_{\mathrm{out}}(g)\big)\, \widehat{T}(\pi, \pi)\, \big(\pi(g) \otimes \rho_{\mathrm{in}}(g)^{-1}\big)\, d\lambda(g), \tag{60}$$

*with $B_\pi \in \mathrm{Hom}_G\big(\pi^* \otimes \rho_{\mathrm{in}},\, \pi^* \otimes \rho_{\mathrm{out}}\big)$. In particular, every equivariant $T$ is block-diagonal across frequencies and acts as the identity on $V_\pi$ and as an intertwiner on the fiber–multiplicity space $V_\pi^* \otimes V$.*

*Sketch.*   Decompose both domain and codomain via Peter–Weyl and write $\widehat{T}$ in blocks $\widehat{T}(\pi, \sigma)$. Conjugation by $(\tau \otimes \rho)$ restricts, on the $(\pi, \pi)$ block, to the representation $\pi^* \otimes \rho_{\mathrm{out}}$ on the codomain multiplicity and $\pi^* \otimes \rho_{\mathrm{in}}$ on the domain multiplicity. Averaging is the orthogonal projection onto the commutant, hence onto $\mathrm{Hom}_G(\pi^* \otimes \rho_{\mathrm{in}}, \pi^* \otimes \rho_{\mathrm{out}})$, and kills $\pi \neq \sigma$ by Schur orthogonality. The displayed formula is the explicit Bochner average of that projection. $\qquad\square$

# D. Implementation Details

In this section, we provide additional information on the implementation details of all of our experiments.

## D.1. Example: Learned $SO(2)$ invariance

**Data generation.** Using polar coordinates $(r, \theta)$, we sample the inner cloud (blue, label $+1$) by drawing $r \sim \text{Unif}[0, 1]$ and $\theta \sim \text{Unif}[0, 2\pi)$, and the outer cloud (red, label $-1$) by drawing $r \sim \text{Unif}[2.3, 3]$ and $\theta \sim \text{Unif}\left[-\frac{\pi}{4}, \frac{\pi}{4}\right)$.

**Feature map and network.** We project inputs $(x, y) \in \mathbb{R}^2$ onto circular harmonics up to degree $M = 4$ with $C = 4$ radial channels as follows: viewing $(x, y)$ as a complex number $z \in \mathbb{C}$ with $r = |z|$ and $\hat{z} = z/r$, define radial basis functions

$$b_n(r) = \exp\left(-\frac{(r - c_n)^2}{2\sigma^2}\right), \quad \sigma = 0.5, \quad c_n \text{ uniform in } [0, 4], \ n = 1, \dots, C.$$

Form the order-$m$ harmonic features by $h^{(m)}(r, \hat{z}) = \left(b_n(r)\,\hat{z}^m\right)_{n=1}^C$ for $m = -M, \dots, M$, and concatenate across $m$ to obtain the embedding

$$H \in \mathbb{C}^{(2M+1) \times C}.$$

We then apply two fully connected complex linear layers

$$L_1 : \mathbb{C}^{(2M+1) \times C} \to \mathbb{C}^{(2M+1) \times C_{\text{hid}}}, \qquad L_2 : \mathbb{C}^{(2M+1) \times C_{\text{hid}}} \to \mathbb{C}^{(2M+1) \times C_{\text{hid}}},$$

followed by an SO(2)-equivariant tensor product:

$$h'_{m_{\text{out}}} = \sum_{m_1 + m_2 = m_{\text{out}}} h_{m_1}\, h_{m_2},$$

with complex multiplication applied channel-wise. We then extract the invariant component $h'_0$ and pass its real part through a final real-valued linear head $L_{\text{final}} : \mathbb{R}^{C_{\text{hid}}} \to \mathbb{R}$ to produce the scalar logit.

We then train a new model for each combination of $\lambda_G, \lambda_\perp$ (see Figure 4) using the Adam optimiser (Kingma & Ba, 2014) for 200 epochs with a learning rate of 0.003. We use a binary cross-entropy loss as task-specific loss.

**Angular perturbation experiment (Figure 5).** To study the interaction between the projection regulariser and violations of exact SO(2) symmetry, we construct a family of "wavey" ring datasets parameterised by an amplitude $\sigma_\perp \geq 0$. For each $\sigma_\perp$ we independently sample angles $\theta_+, \theta_- \sim \text{Unif}[0, 2\pi)$ and define class-conditional radii

$$r_+(\theta_+) = r_{\text{in}} + \sigma_\perp \sin(f\theta_+) + \epsilon_{\text{in}}, \qquad r_-(\theta_-) = r_{\text{out}} + \sigma_\perp \sin(f\theta_-) + \epsilon_{\text{out}},$$

with $(r_{\text{in}}, r_{\text{out}}) = (1.1, 2.2)$, frequency $f = 5$ and independent jitters $\epsilon_{\text{in}} \sim \text{Unif}[-b_{\text{in}}, b_{\text{in}}]$, $\epsilon_{\text{out}} \sim \text{Unif}[-b_{\text{out}}, b_{\text{out}}]$ for $(b_{\text{in}}, b_{\text{out}}) = (0.15, 0.22)$. Mapping $(r_\pm, \theta_\pm)$ to Cartesian coordinates yields two noisy rings labelled $+1$ (inner) and $-1$ (outer). In Figure 8, we consider $\sigma_\perp \in \{0, 0.5, 0.75, 1.0\}$, sample 350 points per class, and split the data into $80\%$ training and $20\%$ test. For each $(\sigma_\perp, \lambda_\perp)$ we then train (i) the approximately $SO(2)$-invariant architecture described above (blue lines), and (ii) a plain real-valued MLP on the raw coordinates (orange).

We see that even for a fixed value of $\lambda_\perp$, the regulariser allows us to capture different effective levels of invariance as the data depart from rotational symmetry; see, for instance, the row with $\lambda_\perp = 1.0$, where the learned classifier remains nearly invariant for small $\sigma_\perp$ and gradually departs from invariance as the angular modulation strengthens. For strongly broken SO(2) symmetry (e.g. $\sigma_\perp = 1.0$), the decision boundary remains "as radially symmetric as possible": away from the perturbed regions the contours revert to circular rings, and in the region between the two classes, around each arm of the star-shaped pattern, the classifier exhibits consistent behaviour across angles.

### D.1.1. SENSITIVITY WITH RESPECT TO $\lambda_G$ AND $\lambda_\perp$

We study the sensitivity of our method to the scalar weights $\lambda_G$ and $\lambda_\perp$ through two ablation experiments. First, we repeat the experiment from Section 4.1 on approximate $SO(2)$ invariance in 2D for $\lambda_G, \lambda_\perp \in \{0, 0.001, 0.01, 0.1\}$; the resulting decision boundaries are shown in Figure 7. When the penalty on the orthogonal component dominates (e.g. $\lambda_\perp = 0.1$ and $\lambda_G \in \{0, 0.001, 0.01\}$), the decision boundary becomes essentially rotationally invariant. In the regime $\lambda_\perp \approx \lambda_G$, the regulariser effectively reduces to standard Tikhonov ($\ell_2$) regularisation and no longer induces a geometric inductive bias. For $\lambda_\perp < \lambda_G$, the learned level sets increasingly depend on angular information.

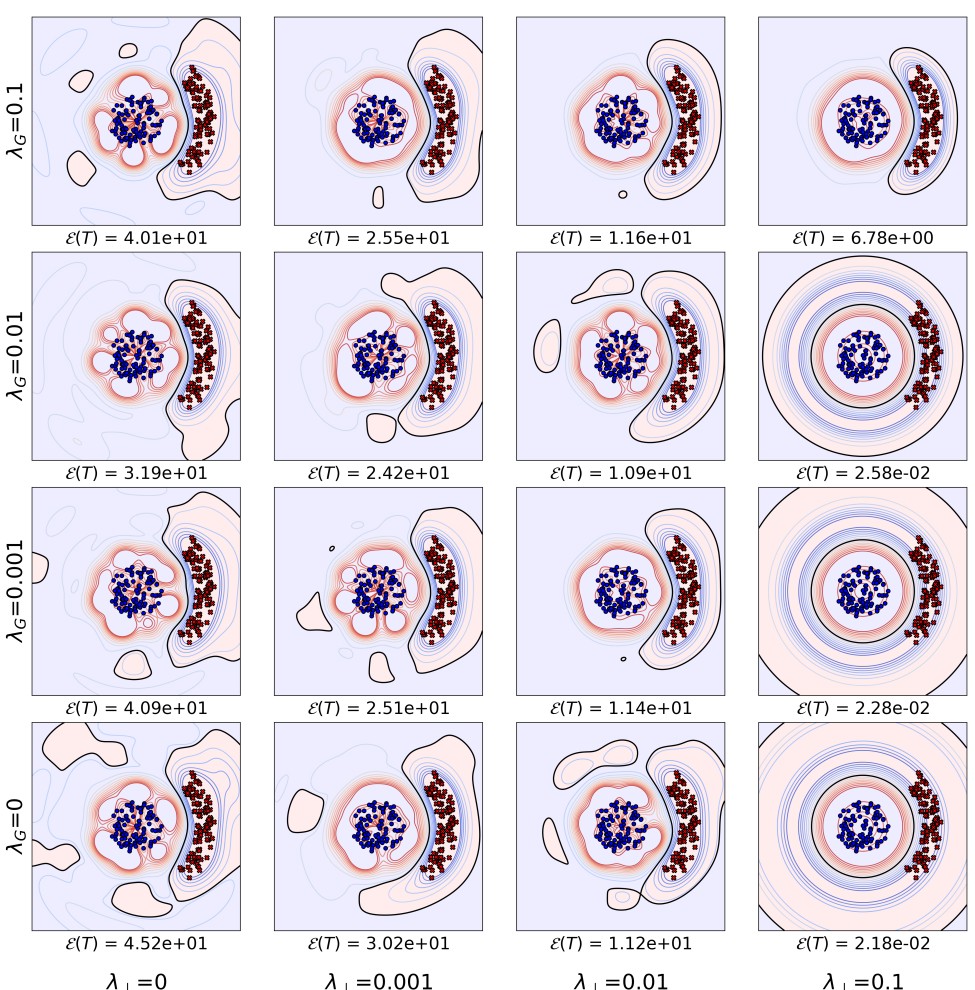

*Figure 7.* Controlling the degree of learned $SO(2)$
invariance by varying the values of $\lambda_G$ and $\lambda_\perp$ over the grid $\{0, 0.001, 0.01, 0.1\}$.

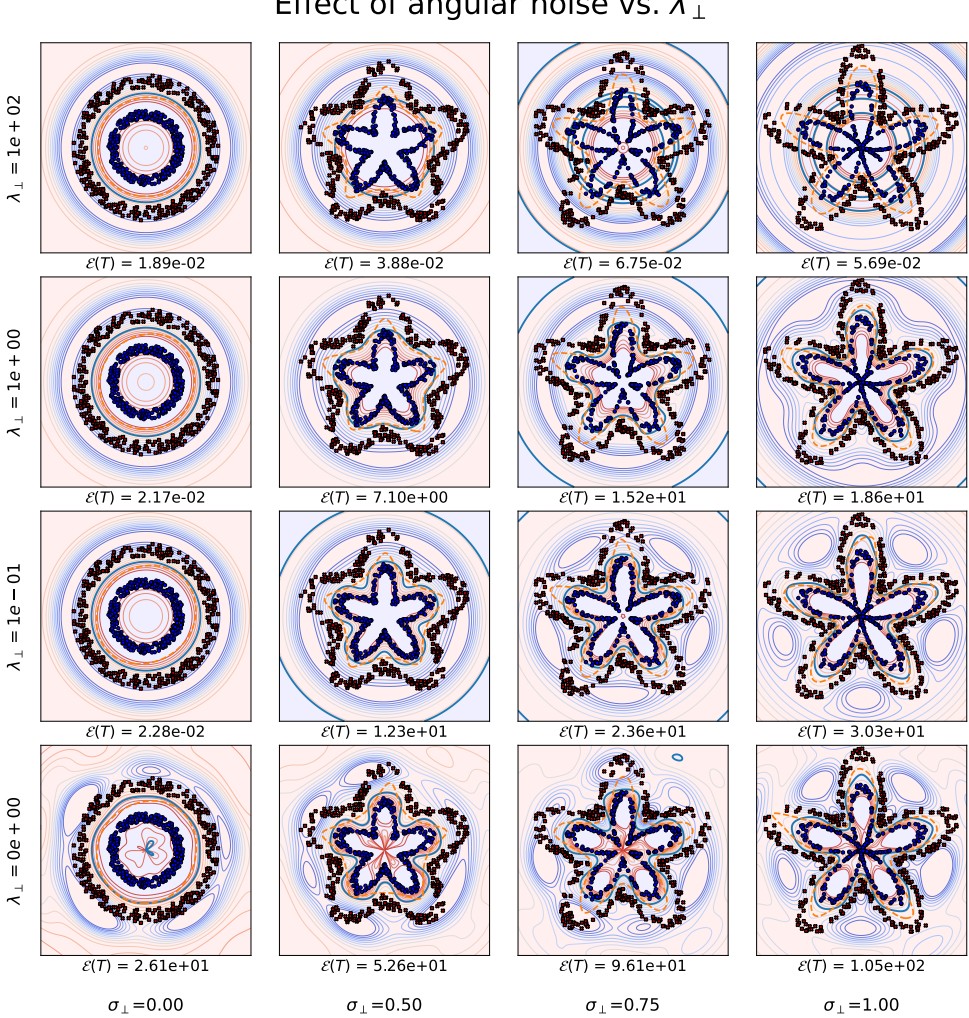

*Figure 8.* Effect of angular perturbations and projection strength. Columns vary the angular wave amplitude $\sigma_\perp$, rows vary the non-equivariant penalty weight $\lambda_\perp$. Blue contours show level sets of the approximately $\mathrm{SO}(2)$-invariant network and points denote training samples. Orange dashed lines are the decision boundary of a non-equivariant MLP. The value $\mathcal{E}(T)$ underneath each panel is the empirical invariance defect, demonstrating that larger $\lambda_\perp$ keeps the classifier close to invariant even as the Bayes decision boundary becomes increasingly angle-dependent.

## D.2. Imperfectly Symmetric Dynamical Systems

For each baseline, relaxed group convolution (`RGroup`) and relaxed steerable CNN (`RSteer`), and for each symmetry setting, we conduct a hyperparameter sweep over learning rate, batch size, hidden width, number of layers, and the number of rollout steps used to compute prediction errors during training, using the same search ranges as Wang et al. (2022c) (see Table 5). We also tune the number of filter banks for group-convolution models and the coefficient for the non-equivariance penalty $\lambda_\perp$ for relaxed weight-sharing models. The input sequence length is fixed to 10. To ensure a fair comparison, we cap the total number of trainable parameters for every model at no more than $10^7$.

*Table 5.* Hyperparameter tuning range for the asymetric smoke simulation data.

| LR | Batch size | Hid-dim | Num-layers | Num-banks | #Steps for Backprop | $\lambda_\perp$ |
|---|---|---|---|---|---|---|
| $10^{-2} \sim 10^{-5}$ | $8 \sim 64$ | $64 \sim 512$ | $3 \sim 6$ | $1 \sim 4$ | $3 \sim 6$ | $0, 10^{-2}, 10^{-4}, 10^{-6}$ |

## D.3. CT Scan Metal Artifact Reduction

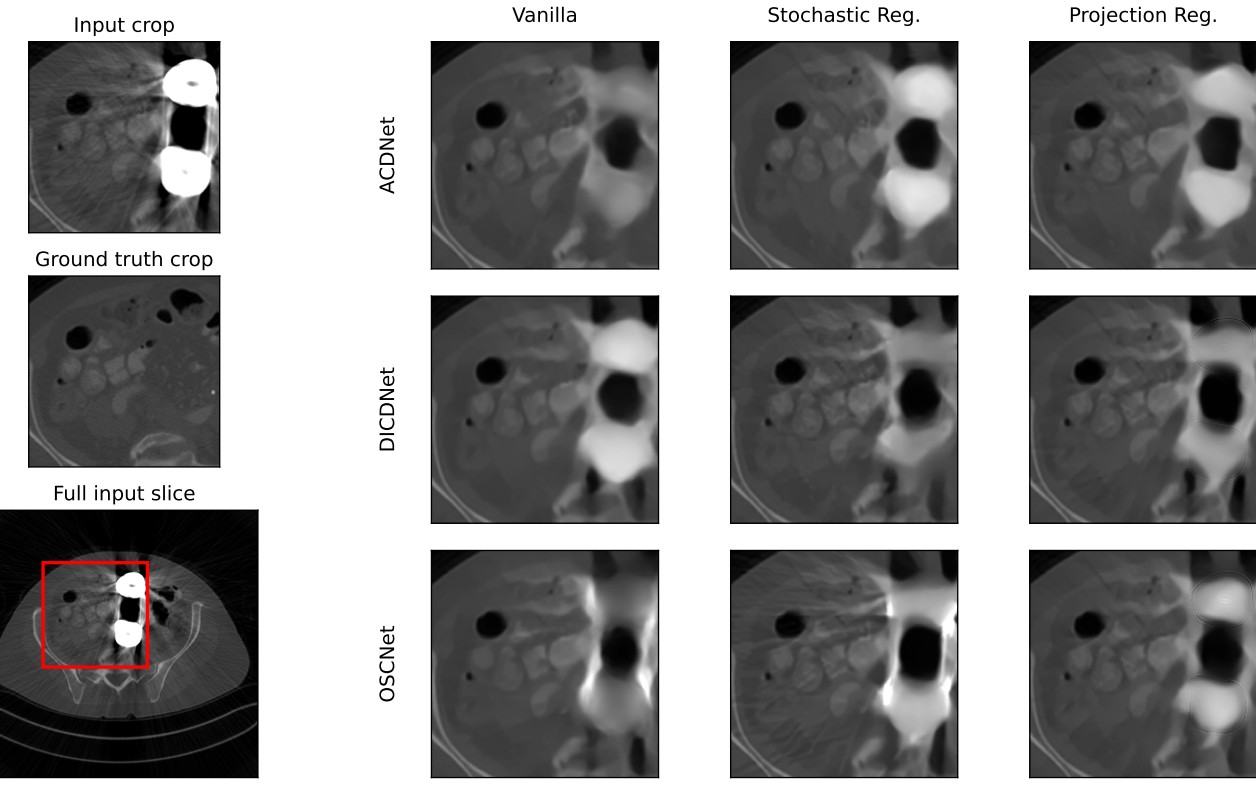

*Figure 9.* Qualitative comparison of the baseline methods (left column) with each of the sample-based (middle column) and our projection-based regulariser (right column) on the metal artefact reduction task. We show a cropped pelvic slice containing two metallic implants that generate artefacts.

### D.3.1. HYPERPARAMETERS

For the most part, we use the same hyperparameters as (Bai et al., 2025). We train for 80 epochs with a batch size of 12 for the baselines and our projection-based regulariser, and a batch size of 4 for the sample-based regulariser. We set the patch size at $256 \times 256$. Optimization uses Adam (Kingma & Ba, 2014) ($\beta_1 = 0.5$, $\beta_2 = 0.999$) with initial learning rate $\eta_0 = 2 \times 10^{-4}$ and a MultiStepLR scheduler (milestones at epochs $\{50, 100, 150, 200\}$, decay factor $\gamma = 0.5$). The model hyperparameters are summarised in Table 6.

The scalar weight for sample-based regulariser is set at $10^6$. To set ours, we performed a hyperparameter sweep over the set $\{1.0, 10^{-1}, \ldots, 10^{-6}\}$ and chose $\lambda_G = 1.0$.

*Table 6.* Hyperparameters for the CT-MAR experiments.

| Parameter | Value |
|---|---|
| $N$ (feature maps) | 8 |
| $N_p$ (concat channels) | 35 |
| $d$ (dict. filters) | 32 |
| Residual blocks / ResNet | 3 |
| Stages $T$ | 10 |

### D.3.2. PROJECTION ONTO THE $C_4$-EQUIVARIANT KERNEL SUBSPACE

We consider steerable CNN layers whose input and output feature spaces are arranged in orientation groups of four (regular-representation channels) for the discrete rotation group $C_4 = \{0, 1, 2, 3\}$ (multiples of $90°$). Let

$$K \in \mathbb{R}^{C'_{\text{out}} \times C'_{\text{in}} \times 4 \times 4 \times s \times s}$$

denote an $s \times s$ convolution kernel with output block index $p \in \{1, \ldots, C'_{\text{out}}\}$, input block index $q \in \{1, \ldots, C'_{\text{in}}\}$, orientation indices $\alpha, \beta \in \{0, 1, 2, 3\}$, and spatial indices $(i, j) \in \{0, \ldots, s-1\}^2$. Let $S$ be the $4 \times 4$ cyclic-shift matrix so that the channel representations of $C_4$ act by $\rho_{\text{out}}(r) = S^r$ and $\rho_{\text{in}}(r) = S^r$ for $r \in \{0, 1, 2, 3\}$. Write $\text{rot}_r$ for rotation of the spatial kernel by $90°r$ (with exact index permutation on the discrete grid).

The natural action of $C_4$ on kernels combines spatial rotation with orientation-channel permutations:

$$\big(\mathcal{A}(r)\, K\big) \;=\; \rho_{\text{out}}(r) \big(\text{rot}_r K\big) \rho_{\text{in}}(r)^{-1} \;=\; S^r \big(\text{rot}_r K\big) S^{-r}. \tag{61}$$

The orthogonal projector onto this subspace is the (finite) Haar average of the action:

$$P(K) \;=\; \tfrac{1}{4} \sum_{r=0}^{3} S^r \big(\text{rot}_r K\big) S^{-r}. \tag{62}$$

Index-wise, for any $(p, q, \alpha, \beta, i, j)$, this reads

$$\big[P(K)\big]_{p,\alpha;\, q,\beta}[i, j] \;=\; \tfrac{1}{4} \sum_{r=0}^{3} \big[\text{rot}_r K\big]_{p,\alpha-r;\, q,\beta-r}[i, j]. \tag{63}$$

Since equation 62 is the average of unitary (permutation + rotation) actions, $P$ is an orthogonal projector: $P^2 = P$ and $P^\top = P$. In practice, equation 62 yields an efficient, exact implementation requiring only four $90°$ rotations and two inexpensive orientation-channel permutations per term.

### D.4. Biomedical Image Classification

#### D.4.1. HYPERPARAMETERS

All baselines (*3D* CNNs, RPPs, and (P)-SCNNs) use the same hyperparameters as in Veefkind & Cesa (2024); see Figure 9 and Tables 12 and 14 in Veefkind & Cesa (2024). In our method, we keep most hyperparameters fixed. We only sweep over the newly introduced parameters $(\lambda_G, \lambda_\perp)$ and tune the learning rate. Table 7 lists the search ranges; we select the best configuration based on validation metrics. Table 8 reports the chosen values.

*Table 7.* Hyperparameter tuning range for our approach on the MedMNIST datasets.

| Learning rate | $\lambda_G$ | $\lambda_\perp$ |
|---|---|---|
| $5 \cdot 10^{-6}, 1 \cdot 10^{-5}, 5 \cdot 10^{-5}, 1 \cdot 10^{-5}, 5 \cdot 10^{-4}$ | $10^{-3}, 10^{-4}$ | $10^0, 10^{-1}, 10^{-2}, 10^{-3}$ |

*Table 8.* Chosen hyperparameters for each configuration on the MedMNIST datasets.

| Parameter | $SO(3)$ | | | $O(3)$ | | |
|---|---|---|---|---|---|---|
| | **Nodule** | **Organ** | **Synapse** | **Nodule** | **Organ** | **Synapse** |
| Learning rate | $5 \cdot 10^{-5}$ | $5 \cdot 10^{-5}$ | $10^{-4}$ | $10^{-4}$ | $10^{-4}$ | $10^{-4}$ |
| $\lambda_G$ | $10^{-3}$ | $10^{-3}$ | $10^{-3}$ | $10^{-4}$ | $10^{-3}$ | $10^{-3}$ |
| $\lambda_\perp$ | $1$ | $10^{-2}$ | $1$ | $10^{-1}$ | $10^{-3}$ | $10^{-1}$ |

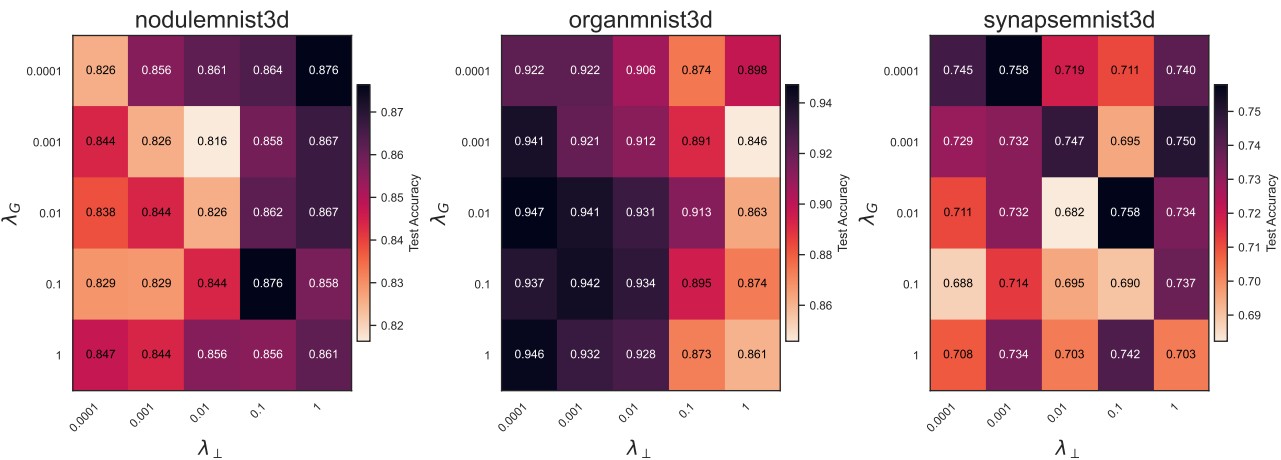

*Figure 10.* Classification accuracy of our approach on the Nodule, Organ, and Synapse tasks of MedMNIST (Yang et al., 2023) while varying $\lambda_G$ and $\lambda_\perp$.

### D.4.2. SENSITIVITY WITH RESPECT TO $\lambda_G$ AND $\lambda_\perp$

We evaluate the sensitivity of our method with respect to the parameters $\lambda_G$ and $\lambda_\perp$. Figure 10 shows performance on the Nodule, Organ, and Synapse tasks of the MedMNIST dataset (Yang et al., 2023) for $\lambda_G, \lambda_\perp \in \{0.0001, 0.001, 0.01, 0.1, 1.0\}$.

We observe that the optimal parameter choice depends on the degree of equivariance present in the task. Nodule and Synapse benefit from larger values of $\lambda_\perp$, which encourage stronger equivariance, whereas Organ achieves the best performance for smaller $\lambda_\perp$. This is expected because Organ requires distinguishing, for example, left from right kidneys or femurs. This is information that is suppressed under perfect rotational or reflection symmetry.

### D.4.3. CONFUSION MATRICES

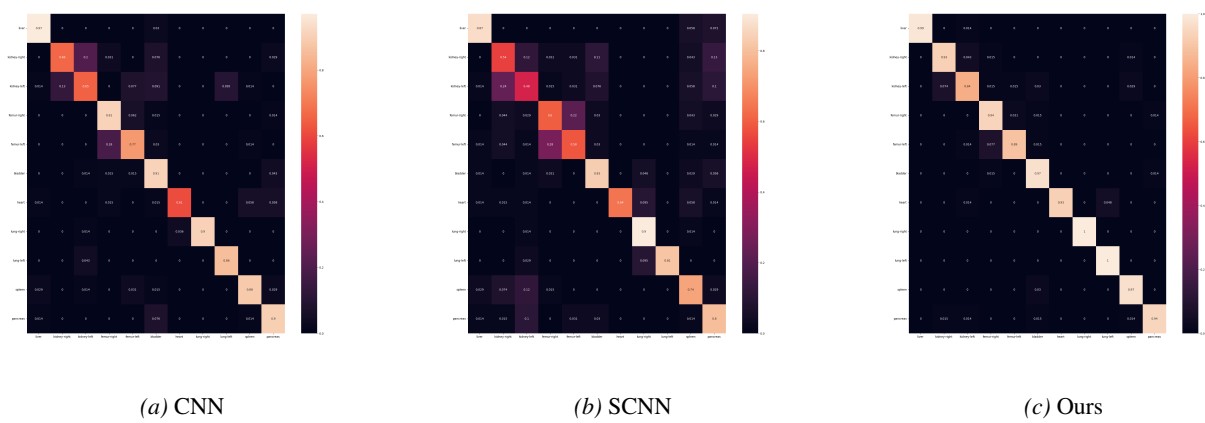

*(a)* CNN        *(b)* SCNN        *(c)* Ours

*Figure 11.* Confusion matrices for a 3D CNN, an SO(3)-equivariant SCNN, and our approach trained on the Organ split of the MedMNIST dataset.

*Table 9.* Results for the CT scan metal artifact reduction task from Section 4.3 for different matrix norms. We report throughput during training and inference as well as total epoch time; the performance metrics are PSNR/SSIM. We consider the spectral, Frobenius (which we use by default in Section 4.3) and infinity norms, as well as the $(p, q)$-norms for $p, q \in \{1, 2, 3\}$.

| Norm | Throughput (no./GPU-s) | | Epoch | AAPM | |
|---|---|---|---|---|---|
| | Train ↑ | Inference ↑ | time (s) ↓ | PSNR ↑ | SSIM ↑ |
| Spectral | 6.59 | 10.16 | 877 | 39.25 | 0.9318 |
| **Frobenius** | 7.22 | 10.11 | 778 | 38.48 | **0.9457** |
| Infinity | 7.73 | 10.12 | 777 | 35.61 | 0.9153 |
| $(1, 1)$ | 7.63 | 10.13 | 785 | 35.57 | 0.8864 |
| $(1, 2)$ | 7.12 | 10.14 | 785 | 38.05 | 0.9365 |
| $(1, 3)$ | 7.12 | 10.13 | 785 | 38.67 | 0.9391 |
| $(2, 1)$ | 7.65 | 10.14 | 783 | 39.33 | 0.9299 |
| $(2, 2)$ | 7.65 | 10.13 | 783 | 38.24 | 0.9430 |
| $(2, 3)$ | 7.61 | 10.14 | 786 | 38.18 | 0.9299 |
| $(3, 1)$ | 7.59 | 10.14 | 787 | 39.10 | 0.9304 |
| $(3, 2)$ | 7.32 | 10.10 | 810 | 39.54 | 0.9322 |
| $(3, 3)$ | 7.18 | 10.16 | 780 | 37.86 | 0.9346 |

We examine the confusion matrices (Figure 11) for a 3D CNN, an SO(3)-equivariant SCNN, and our approach, all trained on the Organ split of the MedMNIST dataset (Yang et al., 2023). The SCNN hard-codes rotational equivariance and therefore tends to confuse left and right femurs and kidneys. The 3D CNN exhibits similar confusion for the kidneys, but distinguishes the femurs more reliably. Our method performs best overall, with substantially fewer left–right confusions.

## E. Sensitivity with respect to norm

In this ablation, we study the impact of the choice of matrix norm in the projection regulariser. We consider the following norms. First, the spectral norm

$$\|A\|_2 = \max_{\|x\|_2=1} \|Ax\|_2, \tag{64}$$

which is equal to the largest singular value of $A$. Second, the Frobenius norm

$$\|A\|_F = \sqrt{\sum_{i,j} a_{i,j}^2}. \tag{65}$$

Third, the (entrywise) infinity norm

$$\|A\|_\infty = \max_{i,j} |a_{i,j}|. \tag{66}$$

Finally, we consider mixed $(p, q)$-norms, defined row-wise as

$$\|A\|_{p,q} = \left( \sum_i \left( \sum_j |a_{i,j}|^p \right)^{\frac{q}{p}} \right)^{\frac{1}{q}}, \tag{67}$$

for $p, q \in \{1, 2, 3\}$. The corresponding results are reported in Table 9. We can see that the choice of norm has only a modest effect on both computational cost and reconstruction quality. Training and inference throughput, as well as epoch time, are nearly identical across all norms, except for the spectral norm, which is about 10–15% slower per epoch, as expected given the need to estimate the largest singular value. In terms of image quality, several choices yield very similar PSNR/SSIM, with the Frobenius and $(p, q)$-norms for $(p, q) \in (2, 2), (1, 3), (3, 3)$ all lying within roughly 1 dB PSNR and 0.01 SSIM of each other. Norms that emphasise elementwise extremal behaviour, such as the infinity norm and the $(1, 1)$-norm, lead to clear degradation in both PSNR and SSIM, indicating that these penalties are too stiff and effectively underfit the reconstruction task. Since the spectral norm brings no systematic performance gains while incurring a noticeable

runtime overhead, and more aggressive entrywise norms harm reconstruction quality, we adopt the Frobenius norm as our default in Section 4.3.

## F. Declaration of LLM use

We used LLMs to aid in the writing process for proof-reading, spell checking, and polishing writing.

