# OpenReview forum: "Approximate Equivariance via Projection-Based Regularisation"
_ICML.cc/2026/Conference — ICML 2026 regular_

### Official Review · Reviewer_MbPV · 2026-02-28

**Soundness:** 3
**Presentation:** 2
**Significance:** 3
**Originality:** 3
**Overall Recommendation:** 5
**Confidence:** 4

**Summary:**

This paper proposes a projection-based regularization framework to enable neural networks to learn (approximate) symmetries from data. To be specific, this framework demonstrates that any linear layer can be decomposed into equivariant and non-equivariant components by the G-smoothing  (Reynolds) operator, and symmetries can be learned by optimizing the norm of the projected parts.
To deal with the projection for continuous symmetry, the authors provide an efficient way to compute this penalty in the Fourier domain, making the model faster and more accurate. Empirical results have shown that this framework has achieved improvement over various methods.

**Compliance With Llm Reviewing Policy:**

Affirmed.

**Final Justification:**

The authors have adequately addressed my main concerns in the rebuttal, and I maintain my evaluation of the paper and keep my original recommendation for acceptance.

**Key Questions For Authors:**

1. Could the author add a discussion about the comparison with [1]?

2. Could the author add [1] as a baseline method in their experiments?



[1] Kim, Hyunsu, et al. "Regularizing towards soft equivariance under mixed symmetries." International Conference on Machine Learning. PMLR, 2023.

**Limitations:**

yes

**Strengths And Weaknesses:**

## Strengths

1. The mathematical framework of this paper is elegant and rigorous, particularly in its use of Fourier analysis to achieve closed-form equivariant projections for continuous groups.

2. An analysis of the extra computation costs is provided, making this work more solid.

3. Extensive and sufficient experiments across diverse domains, including dynamical systems, CT imaging, and medical classification, are provided to demonstrate the efficiency of their methods.

##  Weaknesses

1. In table 1, there are two $\mathrm{+Reg}$ label, which is kind of misleading.

2. In Theorem 3.6 and eq15, the tensor product term $I_{V_\pi} \otimes B_\pi$ feels somewhat abrupt and may confuse readers on how it acts on the signal $f$. An explicit decomposing of $f$ into the direct sum of tensor products  $V_\pi \otimes V_\pi^*$ should be introduced in advance to match eq15.

3. The paper lacks a discussion of an important prior work [1] which also proposes a projection-based equivariance regularizer. A theoretical and empirical comparison with this work is necessary.

[1] Kim, Hyunsu, et al. "Regularizing towards soft equivariance under mixed symmetries." International Conference on Machine Learning. PMLR, 2023.

---

> ### Author Rebuttal · Authors · 2026-03-31
>
> We thank the reviewer for their time invested in reading the paper and for their constructive and helpful review. We address the raised points below.
>
> **[W1]: Double labels**
>
> In Table 1, the proposed regulariser is applied separately to relaxed group convolutional networks (RGroup) and relaxed steerable CNNs (RSteer), and the “+Reg” notation was intended to denote these regularised variants. We agree that this labeling is potentially misleading and will revise Table 1 to distinguish the two variants more clearly in the final version.
>
> **[W2]: Presentation of Theorem 3.6**
>
> Our intent is to use the Peter–Weyl decomposition $L^2(G) \cong \bigoplus\_{\pi\in\widehat G} V\_\pi \otimes  V^\star\_\pi$ so that, after decomposing both the domain and codomain in this basis, an equivariant operator acts blockwise on each isotypic component as $I_{V_\pi}\otimes B_\pi,  B_\pi\in \mathrm{End}(V\_\pi\^\star)$. We agree that this identification should be stated explicitly before Eq. (15), since otherwise it is not immediately clear how the tensor-product term acts on the signal $f$. In the revision, we will therefore add a short paragraph immediately before Theorem 3.6 that introduces the decomposition $L^2(G)\cong\bigoplus_{\pi\in\widehat G} V_\pi\otimes V_\pi^*$, explains that $\widehat T$ correspondingly decomposes into blocks $\widehat T_{\pi\sigma}\in \mathrm{Hom}(V_\sigma\otimes V\_\sigma^\star,\,V_\pi\otimes V_\pi^\star)$, and then states that equivariance forces $\widehat T_{\pi\sigma}=0$ for $\pi\neq \sigma$, while on each diagonal block $\widehat T_{\pi\pi}=I_{V_\pi}\otimes B_\pi$. We will also make Eq. (16) explicit as the induced action on Fourier coefficients, to connect the operator form and the signal-level form more transparently.
>
> **[W3]: Discussion of Kim at al. (2023)**
>
> Thank you for pointing out this relevant reference. We agree that it should be discussed in more detail.
>
> Kim et al. (2023) also regularise toward equivariance via a projection-based penalty, but the setting and scope differ substantially from ours. Their method is designed for mixed symmetries, whereas our focus is on deriving explicit projection penalties for the group actions considered here, including continuous groups where we give a closed-form Fourier-domain construction. While both approaches penalise distance to an equivariant subspace, our paper provides an explicit characterisation and computation of this projection for the studied settings, which is central to both our theory and efficiency claims.
>
> A direct empirical comparison is unfortunately not currently feasible from the information available in their paper. To the best of our understanding, the code is not publicly released, the experiments are conducted (in part) on custom synthetic benchmarks, and the implementation details are not specified at a level that would allow faithful reproduction. In particular, key construction details for the group bases, model architectures, and nonlinear building blocks are not given in sufficient detail for a controlled reimplementation.
>
> There is also a modelling difference in parameterisation: their approach builds on residual pathway parameterisations, whereas our method directly regularises the projection of a layer onto the equivariant subspace. We will add a discussion of Kim et al. (2023) in the related work and clarify both the conceptual connection and the distinction in scope and construction.

---

> > ### Author Rebuttal · Reviewer_MbPV · 2026-04-01
> >
> > The authors have adequately addressed my main concerns in the rebuttal, and I maintain my evaluation of the paper and keep my original recommendation for acceptance.

---

> > > ### Author Response · Authors · 2026-04-08
> > >
> > > We again thank the reviewer for the careful reading, constructive engagement throughout the rebuttal process, and recommendation to accept the paper.
> > >
> > > We are grateful for the helpful comments on the presentation of the mathematical argument in Section 3 and on the discussion of prior work. We believe these points strengthen the paper, and we will incorporate this discussion in the final version.

---

### Official Review · Reviewer_fyyE · 2026-03-11

**Soundness:** 3
**Presentation:** 3
**Significance:** 3
**Originality:** 3
**Overall Recommendation:** 4
**Confidence:** 2

**Summary:**

The paper explains how strictly equivariant networks can be too rigid (and too slow) when real-world symmetries are only approximate see AlphaFold-3 ditching equivariance entirely. This paper regularises toward equivariance instead of hard-coding it. Each layer's weights get orthogonally split into an equivariant part and the rest via the Reynolds operator; penalising the non-equivariant part steers the model toward symmetry without locking it in. For non-discrete groups like SO(n), the projection reduces to zeroing off-diagonal blocks in Fourier space. Unlike sample-based penalties that need extra forward passes per group element, this costs nothing beyond a weight-space operation and has zero estimator variance. Experiments are done on CT artifact reduction and 3D medical classification and show consistent gains over RPP, sample-based penalties, and P-SCNNs, with 40–60% throughput improvements in the CT setting.

**Compliance With Llm Reviewing Policy:**

Affirmed.

**Final Justification:**

rebuttal addressed, no change in recommendation needed

**Key Questions For Authors:**

no questions

**Limitations:**

yes

**Strengths And Weaknesses:**

Soundness:
Solid. The Reynolds-operator decomposition is classical and seems to be correctly applied. Lemma 3.2 gives a tight factor-of-2 equivalence between the projection residual and the worst-case equivariant defect. The per-layer decomposition (Lemma 3.3) properly weights by downstream Lipschitz constants.

Presentation:
Mostly clear, with a clear logical flow from motivation to theory to efficient computation to experiments. Not sure why the Pseudocode in Figure 1 is needed, could that not be just expressen in Formulas? Fig. 2 seems helpful though.

Significance:
Here the applications, namely the CT metal artifact reduction experiment clearly show that this paper is relevant and softly enforced symmetries can have practical advantages

Originality:
While the used tools (The Reynolds operator / group averaging projection, and the observation that equivariant maps are block-diagonal in Fourier space) seem to be well-known in harmonic analysis, the combination is sufficiently new (at least to me) to warrant a "good" here.

---

> ### Author Rebuttal · Authors · 2026-03-31
>
> We thank the reviewer for the careful reading and the positive assessment of our work.
>
> Regarding Figure 1, the pseudocode was intended as an implementation-oriented complement to the formal derivations and to make the paper more accessible to the broader machine learning community. We agree that it is not strictly necessary for the mathematical development, and we will revisit whether this material is better simplified or moved to the appendix in the revised version.
>
> We appreciate the reviewer’s positive comments on the paper’s soundness, originality, and significance.

---

> > ### Author Rebuttal · Reviewer_fyyE · 2026-04-02
> >
> > fullresolved

---

> > > ### Author Response · Authors · 2026-04-08
> > >
> > > We again thank the reviewer for the careful reading and constructive engagement throughout the rebuttal process, and for recommending the paper for acceptance. We are glad that the clarifications regarding the pseudo-code and commutative diagrams addressed your concerns.

---

### Official Review · Reviewer_o7vm · 2026-03-13

**Soundness:** 3
**Presentation:** 4
**Significance:** 3
**Originality:** 3
**Overall Recommendation:** 5
**Confidence:** 4

**Summary:**

This work introduces a way to make neural networks approximately equivariant by projecting each layer onto its equivariant part and penalizing the leftover non-equivariant component during training. The aim is to give a principled middle ground between fully equivariant and fully unconstrained models, with theory linking the penalty to equivariance error and experiments showing efficiency and competitive performance.

**Compliance With Llm Reviewing Policy:**

Affirmed.

**Key Questions For Authors:**

The paper gives a spectral framework for continuous groups such as SO(3), but the experimental evidence for runtime and scalability seems much stronger for the finite-group CT setting than for the continuous-group case. Can the authors provide a more explicit empirical analysis of the computational overhead of the projection step for continuous groups, for example wall-clock cost, scaling with model size, or comparison against alternative approximate-equivariance methods in the SO(3)/O(3) setting?

The method is presented as architecture-agnostic, yet in the conclusion notes the authors mention the penalty term must be derived anew for each model architecture and group action. Could the authors clarify the practical scope of this claim, eg how a practitioner should approach this? In particular, what kinds of architectures admit a straightforward projection derivation? How much manual mathematical or implementation effort is required to apply the method to a different architecture different than the ones studied here?

**Limitations:**

see above

**Strengths And Weaknesses:**

The presentation of the paper is nice. A major strength I found is the core methodological claim, that is, the proposed projection-based regularizer for approximate equivariance, and this is developed concretely in the method section through the projection operator, the equivariant/non-equivariant decomposition, and the training objective. Another strong point is the theoretical justification. The paper does not present the regularizer as a heuristic. Instead, it proves that the norm of the non-equivariant component is tightly related to a natural equivariance-defect measure. The layerwise bounds for deep compositions further strengthen the argument that the proposed penalty is meaningful beyond the single-layer setting. The claim that the method operates at the operator level over the full group orbit, rather than via the pointwise sample-based penalties, is also convincingly supported. On the side of the experiments, the CT artifact reduction experiments provide an efficiency advantage.

However, I think some of the broader claims are only partially supported by the current evidence in this work. The claim that the method consistently outperforms prior approximate-equivariance approaches in model performance is somewhat overstated. The empirical results are generally strong, but not uniformly dominant across every benchmark and setting. In several places the method is competitive rather than clearly superior, so my suggestion is that the wording could be softened.  Also, the claim that the projection can be computed exactly and efficiently in both spatial and spectral domains, especially for continuous groups such as SO(n), is supported in principle, but the practical evidence is narrower than the phrasing suggests. I acknowledge that the mathematical framework is solid, yet the experiments do not fully establish broad efficiency across a wide range of continuous-group architectures and scales. Finally (treat it as a note, I wouldn't ask for more experiments), the claim of robustness across varying symmetry regimes is promising and supported by the MedMNIST experiments, but the scope of evidence is still limited to a relatively small number of tasks and fairly structured symmetry settings.

---

> ### Author Rebuttal · Authors · 2026-03-31
>
> We thank the reviewer for their time invested in reading the paper and for their constructive and helpful review. We address the raised points below.
>
> **[Q1]: Empirical evaluation of runtime and scalability for continuous groups**
>
> We provide an additional empirical analysis for the continuous-group experiments in Sections 4.1 and 4.4:
>
> For the $SO(2)$ experiment in Section 4.1, we fix $\lambda=1$ and vary the hidden dimension $d$. Table 1 shows that the projection penalty accounts for $23\\%$-$30\\%$ of the total training step, consistent with the analysis in Section 3.5: both the dense forward pass and the projection penalty scale linearly with the number of parameters, so the overhead remains approximately constant.
>
> For the $SO(3)$ experiments in Section 4.4, we vary the width multiplier $c$ and study parameter count, training and inference throughput (Tables 2-4). Across tested widths, our method achieves higher inference throughput than both RPP and the $SO(3)$ SCNN, and higher training throughput than RPP at all widths and than the $SO(3)$ SCNN at $c=1,2$.
>
> In this setting, ESCNN parameterises models via spatial kernels, and the projection is realised as a dense linear map on convolution kernels. Caching this map becomes prohibitively expensive for $c>3$, and the projection overhead grows from $16.7\\%$ at $c=1$ to $33.7\\%$ at $c=2$ and $59.8\\%$ at $c=3$. We therefore separate the analysis into a matched-scaling study (Tables 2-4) and the operating-point comparison used in the main experiment (Table 5). To match Veefkind and Cesa, we keep the baselines at $c=6$, while our current implementation is limited to smaller widths ($c=2$). At these operating points, our model still performs best on 2 of 3 datasets, while also achieving substantially higher training and inference throughput than RPP and the $SO(3)$ SCNN.
>
> Table 1. Train time and projection-penalty overhead vs. hidden dimension d (Sec. 4.1)
> |d|Train(ms)|Penalty(ms)|Share(%)|
> |---|---|---|---|
> |1024|5.594|1.671|29.88%|
> |2048|15.984|3.716|23.25%|
> |4096|54.620|13.231|24.22%|
> |8192|205.564|51.298|24.95%|
>
> Table 2. Parameter count vs. channels c (Sec. 4.4)
> |Model|c=1|c=2|c=3|
> |---|---|---|---|
> |CNN|520k|2.03M|4.56M|
> |SO(3) SCNN|4k|15k|33k|
> |P-SCNN|310k|1.23M|2.7M|
> |RPP|520k|2.05M|4.59M|
> |Ours|520k|2.03M|4.56M|
>
> Table 3. Training throughput (k samples/s) vs. channels c (Sec. 4.4)
> |Model|c=1|c=2|c=3|
> |---|---|---|---|
> |CNN|1.99|1.34|0.78|
> |SO(3) SCNN|1.00|0.60|0.42|
> |P-SCNN|0.13|0.12|0.11|
> |RPP|0.69|0.43|0.28|
> |Ours|1.34|0.64|0.21|
>
> Table 4. Inference throughput (k samples/s) vs. channels c (Sec. 4.4)
> |Model|c=1|c=2|c=3|
> |---|---|---|---|
> |CNN|9.34|5.09|2.37|
> |SO(3) SCNN|6.62|3.52|1.78|
> |P-SCNN|6.58|3.51|1.77|
> |RPP|4.11|2.20|1.05|
> |Ours|10.18|5.61|2.53|
>
> Table 5. Metric comparison of models in Sec. 4.4
> |Model|c|Params|Train(samples/s)|Infer(samples/s)|Epoch(s)|
> |---|---|---|---|---|---|
> |CNN|6|18.14M|365.28|897.08|3.17|
> |SO(3) SCNN|6|128k|196.70|666.67|5.890|
> |P-SCNN|6|11.09M|82.44|665.92|14.05|
> |RPP|6|18.26M|130.72|391.56|9.06|
> |Ours|2|2.03M|638.15|5606.52|1.816|
>
> Table 6. Projection-penalty overhead vs. channels c (Sec. 4.4)
> |Model|c|Params|Train(samples/s)|Infer(samples/s)|Epoch(s)|Overhead|
> |---|---|---|---|---|---|---|
> |Ours|1|516k|1339.73|10177.31|0.865|16.72%|
> |Ours|2|2.03M|638.15|5606.52|1.816|33.68%|
> |Ours|3|4.56M|209.38|2526.24|5.534|59.79%|
>
> **[Q2]: Practical Considerations for Implementation in other Architectures**
>
> By “architecture-agnostic,” we meant the general recipe of using the projection as a regulariser toward equivariance, not a zero-effort drop-in implementation for arbitrary architectures. In practice, the main challenge is deriving the projection in closed form. As discussed in Section 3, this is most straightforward for learnable linear operators, such as dense layers and convolutional kernels, once the relevant group representations are specified. The main difficulty arises for nonlinearities and architecture-specific modules, for which similarly simple closed-form projections are generally not available.
>
> A natural practical route is to start from an equivariant architecture, relax its learnable linear layers to allow non-equivariant parameters, and then apply the projection machinery from Section 3 to those layers while keeping the remaining equivariant components unchanged.
>
> We will revise the conclusion to make this practical scope more explicit.
>
> **[W1]: Wording too strong**
>
> We agree that the wording was too strong in a few places, and we will soften those claims in the revision.
>
> *Abstract*
>
> "In our experiments, our method is competitive with prior approximate-equivariance approaches in task performance, while achieving substantial runtime gains over sample-based regularisers."
>
> *Lines 050-053*
>
> "We empirically show that our method is competitive with existing approximate-equivariance approaches in task performance, while offering especially large runtime gains over sample-based regularisers."

---

> > ### Author Rebuttal · Reviewer_o7vm · 2026-04-01
> >
> > Thank you the effort and for addressing my concerns. I adjust my score to acceptance.

---

> > > ### Author Response · Authors · 2026-04-08
> > >
> > > We again thank the reviewer for the careful reading and constructive engagement throughout the rebuttal process, and for increasing their score.
> > >
> > > We are glad that the clarifications regarding the runtime evaluation in the continuous-symmetry experiments and the practical aspects of our work addressed your concerns. We believe these points strengthen the paper, and we will include this discussion in the final version.

---

### Official Review · Reviewer_ttC1 · 2026-03-13

**Soundness:** 4
**Presentation:** 2
**Significance:** 3
**Originality:** 3
**Overall Recommendation:** 4
**Confidence:** 4

**Summary:**

The paper provides a theoretically grounded framework for training approximately equivariant networks, with the approximate equivariance being encouraged on the layer level through a particular projection operator. For this purpose, a projection operation is derived which splits the operator orthogonally into an equivariant component and a residual non-equivariant component. For infinite (continuous) groups such as SO(n), the projection can be performed efficiently in the Fourier domain due to the fact that equivariant operators are block diagonal there, but for finite groups it is also practically possible to compute it directly in the signal domain.

**Compliance With Llm Reviewing Policy:**

Affirmed.

**Final Justification:**

The rebuttal has reinforced my original positive assessment of the paper. I recommend it for acceptance.

**Key Questions For Authors:**

1. Two of the four tasks (Sections 4.1 and 4.4) test invariance rather than general equivariance. How can you verify that the full expressive power is retained when the task only evaluates invariance?


2. I have no prior experience with CT-scan metal artefact reduction, and I thus have no intuition about the utility of addressing this as an equivariant task. Why is equivariance wrt the group of multiples-of-90-degrees rotations a relevant/suitable inductive bias for this task?

**Limitations:**

yes

**Strengths And Weaknesses:**

The main contribution, in my opinion, is the derivation of the general theoretical framework to evaluate the projection operator. The related idea of studying the operator represented by a layer in terms of orthogonal bases has been explored in the past (e.g.  Bansal et al., "Can We Gain More from Orthogonality Regularizations in Training Deep CNNs?", NeurIPS'18),  but the present paper makes the crucial connection to split the operator orthogonally into the equivariant and the residual parts (in contrast to earlier work, where the orthogonal decomposition was not informed by the operator).

The mathematical machinery behind this is well-established, but the application of it in this particular setting is clever, elegant, theoretically and technically sound. The derivations are based on reasonable assumptions (for example, the Lipschitz assumption in Sec. 3.1 should be ok except for very exotic choices of activation functions) and are justified by established theory and relevant references.

While the writing is for the most part clear, the logical link between some sections and paragraphs is sometimes vague and unclear, which results in a somewhat disorganised impression (the most prominent case of this is in section 2, which just lists a series of concepts -- some of which are not even directly referenced later -- with no overarching narrative connecting them; another example is at 095-096 where Zhong et al. vaguely "apply related ideas").

Apart from the presentation, the main weakness is the limited experiments and selection of evaluation tasks (two are invariant, which is a more restricted form of equivariance).

---

> ### Author Rebuttal · Authors · 2026-03-31
>
> We thank the reviewer for their constructive and helpful review. We address the raised points and questions below.
>
> **[Q1]: Equivariance in Invariant Task**
>
> We agree that invariant-task accuracy alone does not show that the learned representation remains equivariant.
>
> However, while the final classifier is only required to be (approximately) invariant, the intermediate layers are regularised toward equivariance, with only the final readout collapsing to the scalar channel. Denoting the model up to the penultimate layer by $T$, we can assess whether this structure is reflected in the learned representation by also measuring the empirical relative equivariance error:
> $$
> \mathcal{E}\_{\mathrm{equiv,rel}}=\mathbb{E}\_{g\in G,\;x\in X}\left[\frac{\\|T(g\cdot x)-\rho(g)T(x)\\|}{\\|x\\|}\right].
> $$
>
> In Tables 1-3, we see that in Sec. 4.1, increasing λ reduces the penultimate-layer equivariance error from 0.93 to 1.41e-4, while also reducing the final invariance error from 0.78 to 3.67e-5. We observe the same trend in Sec. 4.4. In the 3D setting, continuous rotations are affected by voxel discretisation, so we additionally evaluate the discrete subgroup of 24 right-angle rotations; there, the strictly equivariant baseline achieves near-zero error and our regularised model again improves substantially with larger λ. We will revise the text to make this distinction explicit: the invariant tasks do not by themselves characterise full equivariant expressive power, but the hidden-representation analysis shows that the regulariser makes the learned representation measurably more equivariant.
>
> Table 1. Rel. L2 eq./inv. error under continuous $SO(2)$ rotations (Sec. 4.1)
> |λ|M|Err|
> |---|---|---|
> |0.0|Inv|0.78|
> ||Eq|0.93|
> |0.001|Inv|0.53|
> ||Eq|0.37|
> |0.01|Inv|3.67e-05|
> ||Eq|1.41e-04|
> |1.0|Inv|3.36e-05|
> ||Eq|1.34e-04|
>
> Table 2. Rel. L2 eq./inv. error under continuous $SO(3)$ rotations (Sec. 4.4)
> |Model|M|Nod|Syn|Org|
> |---|---|---|---|---|
> |3D CNN|Inv|0.96|1.39|1.08|
> ||Eq|N/A|N/A|N/A|
> |Ours (λ=0.0)|Inv|0.80|1.47|1.01|
> ||Eq|0.51|0.80|0.73|
> |Ours (λ=1.0)|Inv|0.19|1.05|0.73|
> ||Eq|0.14|0.65|0.66|
> |SO(3) SCNN|Inv|0.38|0.80|0.69|
> ||Eq|0.14|0.43|0.58|
>
> Table 3. Rel. L2 eq./inv. error under $24$ discrete $90°$ rotations (Sec. 4.4)
> |Model|M|Nod|Syn|Org|
> |---|---|---|---|---|
> |3D CNN|Inv|0.75|1.39|1.03|
> ||Eq|N/A|N/A|N/A|
> |Ours (λ=0.0)|Inv|0.73|1.36|0.99|
> ||Eq|0.44|0.75|0.73|
> |Ours (λ=1.0)|Inv|0.12|0.55|0.56|
> ||Eq|0.07|0.23|0.42|
> |SO(3) SCNN|Inv|1.03e-04|2.99e-04|2.08e-04|
> ||Eq|4.74e-05|1.70e-04|2.05e-04|
>
> **[Q2]: Rotational equivariance in CT MAR**
>
> The physical process underlying CT streak artifacts is, in the ideal continuous setting, equivariant with respect to planar rotations in $SO(2)$: rotating the scan geometry and object rotates the resulting streak artifacts accordingly. In reconstructed images, this symmetry is only approximate, because discretisation, interpolation, and measurement effects break exact rotational symmetry. This makes CT MAR a natural setting for approximate rather than strict equivariance.
>
> Following Bai et al. (2025), we restrict the implementation to the finite subgroup $p4$ of rotations by multiples of $90^\circ$. This is particularly suitable for discretised image data because these rotations preserve the pixel grid and therefore avoid introducing interpolation artefacts. It also allows a direct comparison under the same assumptions as prior work.
>
> **[W1]: Presentation**
>
> We appreciate this feedback and agree that Section 2 previously read too much like a list of concepts without a sufficiently explicit narrative. We have revised the section to improve the logical flow and to clarify how the cited prior work connects to our formulation.
>
> In particular, lines 092–096 now read:
> “A number of works penalise pointwise deviations from equivariance constraints using randomly sampled group transformations: Bai et al. (2025) use this strategy for artifact reduction in imaging, Kouzelis et al. (2025) for VAEs in generative modelling, and Zhong et al. (2023) for depth and normal prediction in images.”
>
> We will include this revision in the final manuscript.

---

> > ### Author Rebuttal · Reviewer_ttC1 · 2026-04-01
> >
> > I thank the authors for the detailed rebuttal.
> > Provided that the authors include the rebuttal in their updated manuscript, I retain my assessment of the paper and recommend it for acceptance with my original score

---

> > > ### Author Response · Authors · 2026-04-08
> > >
> > > We again thank the reviewer for the careful reading and constructive engagement throughout the rebuttal process.
> > >
> > > We are glad that the clarifications on approximate equivariance in the latent representations and $p4$ symmetry in metal artifact reduction addressed your concerns. We will include this discussion in the final version of the paper.

---

### Decision · Program_Chairs · 2026-04-30

**Decision:**

Accept (regular)

**Comment:**

The paper presents a method for training approximately equivariant models using a projection-based which decomposes a layer into equivariant and non-equivariant components.  Reviewers generally agreed the theoretical derivation made for an elegant and sound method. Reviewers were mixed on the writing and scope of the experimental evidence but were generally positive and most concerns were addressed by the rebuttal.  In particular, the authors agreed to soften some claims.  Overall, the method provides a useful entry into the space of equivariant and approximately equivariant methods seeking to leverage symmetry without the potential efficiency and overconstraint issues.